# Morphological response to climate-induced flood-event variability in a subarctic river

**Linnea Blåfield[1],** Carlos Gonzales-Inca[1], Petteri Alho[1,2], Elina Kasvi[1]

[1]Department of Geography and Geology, University of Turku, Finland

[2]Finnish Geospatial Research Institute FGI, National Land Survey of Finland, Espoo, Finland

Correspondence to: Linnea Blåfield, linnea.m.blafield@utu.fi

Keywords: Sediment transport hysteresis, Computational modelling, Flood sequencing, Hydroclimatics

Highlights:

- Sediment transport hysteresis pattern is dependent on the number and volume of flood peak sequences

- Flood-event type significantly impacts the rivers morphological response

- Increase of multi-peaking flood-events, mean temperature, and changing precipitation patterns affects the future river system stability

- Hydrograph shape can be associated to specific preceding climatic conditions

Abstract

This study examined the effects of climate-induced flood-event variability and peak sequencing on morphological response and sediment transport hysteresis patterns in a subarctic river. We classified 32 years of discharge hydrographs from a subarctic river according to their spring flood hydrograph shapes and peak sequences. These classified flood-event types and their frequencies were statistically analysed against seasonal and annual climatic conditions from the corresponding time periods. Morphodynamic modelling was employed to examine the effects of flood-event hydrograph shape and sequencing on morphological response and sediment transport hysteresis patterns during floods. The findings highlight the critical role that hydrograph shape and sequencing play in influencing river morphology and sediment transport dynamics, as each flood-event type produced distinct sediment transport hysteresis patterns and morphological outcomes. Double-peaking floods resulted in relatively more heterogeneous and complex morphological outcome compared to single-peaking floods. Variance and trend analyses revealed that prevailing climatic conditions significantly influence the hydrograph shapes of spring flood

events. Annual mean temperature, total precipitation, and snow accumulation, together with cold season mean temperature, spring rainfall, and May cumulative temperature, had the greatest impact on the type of spring flood event observed. Significant increasing trends were identified in annual and spring mean temperatures, spring rainfall, and the frequency of rain-on-snow events. This suggests that ongoing climatic shifts are actively modifying the nature of spring flood events, favouring more complex and variable hydrograph forms. Consequently, future sediment transport and morphological evolution in subarctic rivers are likely to become increasingly event-driven, less predictable, and more sensitive to interannual climatic variability. These changes emphasise the need for adaptive management strategies that can accommodate the emerging hydrological and geomorphological dynamics under a changing climate.

## 1. Introduction

Hydrological variability significantly affects riverine sediment fluxes, especially in cold climate rivers where sediment transport is highly seasonal, occurring predominantly during spring floods (Syvitski, 2002; Favaro & Lamoureux, 2015; Zhang et al., 2022). Snowmelt-driven spring floods carry majority of the annual sediment budget and therefore, they define the timing and volume of sediment transport and ultimately the whole river morphology. Currently, cold climate rivers are experiencing rapid shifts in hydroclimatic conditions, influencing the flow-sediment interaction in the river systems (Meriö et al., 2019; Beel et al., 2021; Li et al., 2021; Zhang et al., 2023; Blåfield et al., 2024a). As hydroclimatic conditions evolve, the characteristics of flood-events are also changing with implications to the traditional sediment transport dynamics. For instance, the shift in the snow-to-precipitation ratio and changes in the timing and intensity of snowmelt have already altered flood hydrographs i.e., the shape, magnitude, duration, and sediment transport capacity of events in cold-climate rivers (Wohl et al., 2017; Gohari et al., 2021; Hopwood et al., 2021; Zhang et al., 2022; Blåfield et al., 2024a; Lintunen et al., 2024). Flood-events are usually classified by their generating processes (e.g., intense precipitation, snowmelt, rain-on-snow, ice jamming, dam break etc.), with less emphasis on the event type and sequences itself. Previous studies (Viglione et al., 2010; Fischer et al., 2019; Gohari et al., 2022), however, have reported that ongoing regime shifts have altered flood-event shapes. Over the past century, multi-peaking floods have become more common, not only in central Europe, but also in high-latitude regions.

In multi-peaking floods, the order and duration of different peaks significantly affects the sediment transport volume and the pattern of sediment transport hysteresis because the flow conditions control when, how much, and what type of sediment is mobilised, reworked, or deposited within the river system (Mao, 2018). Therefore, understanding the contribution of flood-event sequences to sediment transport is crucial for predicting the impact of climate change on fluvial sediment dynamics and the morphological response of river systems (Mao, 2012; Karimaee Tabarestani & Zarrati, 2015). This is particularly important in cold-climate rivers, which have historically experienced a single major snowmelt-driven flood and low sediment loads. However, due to hydroclimatic regime shifts, altered fluvial dynamics and possible permafrost or glacier melt, these regions are increasingly becoming hotspots for

elevated sediment transport (Syvitski et al., 2002; Li et al. 2021; Zhang et al., 2022). Recent studies indicate that migration rates of large, sinuous rivers in the Arctic permafrost region have slowed by 20% during the past 50 years due to decreased fluvial energy and increased bank shrubification (Ielpi et al., 2023). Contrasting findings have been made on the Tibetan Plateau where migration rates have increased by 34% due to increased discharge volumes and river bank destabilisation caused by permafrost melt (Sha et al., 2025). In boreal-subarctic regions, where the focus of this study is, the fluvial activity and extreme discharge events outside the spring flood season have increased while spring flood peaks have decreased significantly (Korhonen & Kuusisto, 2010; Lintunen et al., 2024). The increased fluvial activity outside traditional flood season is caused by increasing number of extreme rainfall events (Nikulin et al., 2011) which are intensifying bank erosion and sediment transport (Kärkkäinen & Lotsari, 2022). However, the annual total volume of water has not yet changed (Lintunen et al., 2024). All these findings suggest that climate change has diverse impacts on fluvial dynamics across the high-latitude region, and therefore more focus should be paid sediment transport dynamics and the hysteresis pattern under evolving discharge conditions. Understanding these processes is essential because sediment transport not only shapes river morphology but also governs aquatic habitats, influences nutrient fluxes, and affects infrastructure stability.

One effective way to evaluate the sediment transport process and morphological response of the river channel is through analysis of sediment transport hysteresis patterns, which reflect the sediment transport affected by riverbed structure, sediment composition and availability at different stages of the flow hydrograph (Williams, 1989; Reesink & Bridge, 2011; Gunsolus & Binns, 2017). In cold climate rivers various types of sediment transport hysteresis have been observed due to highly seasonal and varying sediment availability between catchments (Vatne et al., 2008; Kociuba, 2021; Wenng et al., 2021; Zhang et al., 2021; Liébault et al., 2022). Yet, measuring bedload and hysteresis in natural rivers during high flows remains challenging and is prone to biases. As a result, long time series of bedload transport and hysteresis are scarce worldwide (Mao, 2018; Zhang et al., 2023). Thus, we rely on laboratory experiments, computational modelling, and field measurements of suspended load when evaluating and measuring the current, and predicting the future sediment fluxes and morphodynamic response of the river channels.

The ability to evaluate and predict the effects of climate change on sediment transport rates and morphological response is essential not only for understanding fluvial morphodynamics, such as channel stability and sediment connectivity but also for a wide range of river engineering and management applications (Mao, 2018; Gupta et al., 2022; Najafi et al., 2021). Therefore, this study aims to: i) Analyse and classify the variation in flood-event hydrographs over the past 32-years in a subarctic river, ii) Link the flood-events to seasonal and annual climate conditions, and iii) Evaluate the channels morphological response distinctive to each flood-event type utilising morphodynamic modelling and sediment transport hysteresis analysis. We expect to detect linkages between the flood-event hydrograph shape and climatic conditions as well as diverse patterns of morphological response and sediment transport hysteresis. The study was conducted on a river reach in

Finland, located at 70° North latitude. Despite its high latitude, Finland has a relatively mild climate compared to other regions at similar latitudes, such as Siberia, northern Canada, and Alaska, largely due to the warming influence of the Gulf Stream and the North Atlantic Drift. As a result, Finland is mostly free of permafrost (Luoto et al., 2004), although small areas of permafrost exist in the form of palsa mires. These palsas are primarily found in north-western Finland (Seppälä, 1997; Gisnås et al., 2017; Verdonen et al., 2023). Nevertheless, Finland experiences seasonally frozen ground for periods ranging from four (South) to eight (North) months each year (Rimali, 2019).

## 2. Study area

The meandering and unregulated Pulmanki River locates in northern Finland (Fig. 1A)The river is a tributary to Tana River which flows into the Arctic Ocean on Norwegian side of the border (Fig. 1A). The river is divided into two separate sections by the Lake Pulmankijärvi (Fig. 1B). The area of interest in this study is a 6-kilometre-long reach on the upper course of the Pulmanki River approximately 13 meters above the mean sea level (a.m.s.l) (Fig. 1B). This reach consists of 13 meander bends with a reach sinuosity of 2.4. The bankfull width of the river varies between 60 to 100 metres, depending on the valley confinement. The river flows through glaciolacustrine and glacio-fluvial sediments deposited on the fjord bottom after the final wasting of Fennoscandian ice sheet (Mansikkaniemi,1967; Hirvas et al., 1988; Johansson et al., 2007). The D50 value of the channel bed material ranges from 0.1 mm to 4 mm and a sandy bedload (D50 0.43 mm) dominates the sediment transport. The amount of suspended material is minimal (0-180 mg/L), even during the spring flood (Lotsari et al., 2020). The bed morphology is typical for sand bed rivers and consists of dunes, ripples, pools, and riffles, the bed is unvegetated and mobile through the year. The channel is frozen from October to May, and the seasonal discharge ranges from 0.5 to 100 m$^3$/s. A spring flood generated by the snowmelt occurs annually in late-May or early June. Lower discharge peaks are associated with precipitation events during July, August and September. The river belongs to subarctic-nival hydrological regime (Lininger and Wohl, 2009) and to Köppen climate class: "Cold, without dry season, but with cold summer" as the area is affected by the great Asian continent and both the Atlantic Ocean and the Gulf Stream. Based on the Nordic permafrost model by Gisnås et al., (2017) majority of the catchment is permafrost free (Fig. 1C). The south-western corner has 10-50 % probability of sporadic permafrost according to the model results based on land cover, snow accumulation and temperature data. However, no confirmed field observations of sporadic permafrost from the area exist, and therefore we consider this catchment and river system as non-permafrost river.

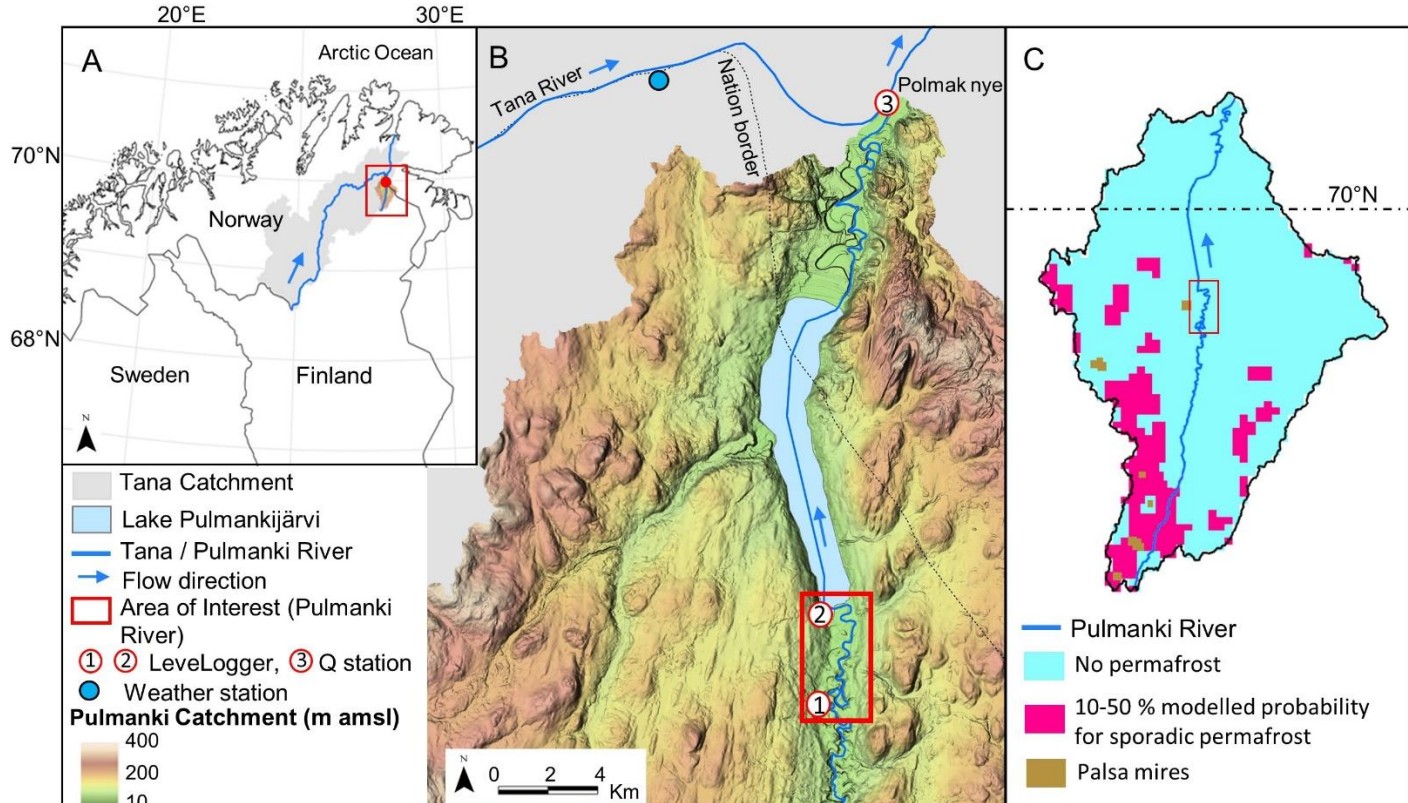

Figure 1. Area of interest. A) The study area's location in the Northern most Finland. B) Model area is marked with rectangle, and the locations of LeveLogger sensors (LL), discharge (Q), and weather station with circular markers. C)The probability of sporadic permafrost within the catchment based on the Nordic permafrost model by Gisnås et al., 2017. Pulmanki catchment 2x2 m DEM by National Land Survey of Finland.

## 3. Data & Methods

Discharge hydrographs of the years 1992-2023 were analysed and classified to recognise variability in spring flood-event shapes. The most typical flood-event of each hydrograph type was selected for morphodynamic modelling to evaluate the channels morphodynamic response and sediment transport dynamics. The flood-events extracted from the classified hydrographs were linked with climate data from equivalent time period to examine possible connections between climate and flood-event shapes. Mann-Kendall trend test was run on the hydroclimatic variables to detect possible trends in the time-series. Continuous discharge and water level monitoring has been conducted in Pulmanki River since 2008 during open water season (May-September). The Pulmanki River discharge time-series was complemented with Polmak discharge station data from Tana River (Fig. 1) to cover the whole 32-year time period. Sediment and bedload transport samples were collected during the spring and autumn field campaigns in 2019 from various discharge conditions.

## 3.1 Hydrograph measurements and generation

Hydrographs of open water season were generated utilising a combination of data sources. For the years 2008-2023, rating curves based on a combination of field data were generated: water pressure sensor data (Levelogger 5, Solinst), water level data measured with Virtual Reference Station-Global Navigation Satellite System (VRS-GNSS), and discharge data measured with Acoustic Doppler Current Profiler (ADCP M9, Sontek). Each year, the water pressure sensors were placed into the upper Pulmanki River after ice-breakup in spring and picked up before winter (see locations in Fig 1). This way the sensors covered the whole open water season and seasonal variations of water pressure, water level and discharge with 15 minute intervals. The location of the sensors was identical each year. To compensate atmospheric influence on water pressure, an air pressure sensor data from Solinst Barologger was subtracted from the water pressure readings. During field campaigns in May and September water level and discharge were measured daily from the LeveLogger locations for creating rating curves between LeveLogger pressure, water level (WL) and discharge (Q). Based on the rating curves, a 3$^{rd}$ order polynomial function was selected for calculating annual hydrographs of open water seasons (Figure 2A).

For the years 1992-2007, openly available daily discharge data from Polmak measurement station, maintained by the Norwegian Water Resources and Energy Directorate (NVE) was used. The station is located in the main channel of Tana River at the spot where Pulmanki River discharges into Tana (see Fig. 1, Q station number 3), and has been operating since November 1991. The discharge for Pulmanki River was derived from the Polmak station data using rating curve and 3$^{rd}$ order polynomial function between the Polmak station discharge (Q) and Pulmanki River Q of 2008-2023 derived from the LeveLoggers (Figure 2B). Note that the discharge of the Tana River continues to rise even as the discharge of the Pulmanki River decreases, owing to the fact that the Tana River drains a catchment area 20 times larger than that of the Pulmanki River. The Pulmanki River catchment (a sub-catchment of the Tana) is situated approximately 200 km downstream from the Tana River headwaters and lies in much lower terrain (with a maximum elevation of 400 m above sea level), compared to the Tana catchment, which includes areas reaching up to 1,100 m above sea level. This results in a delay in snowmelt, causing peak runoff in the Tana River to occur later than in the Pulmanki catchment.The final hydrographs of Pulmanki river are based on these two equations and data sources. The hydrographs were validated against ADCP discharge measurements from Pulmanki River main channel. These measurements were excluded from the rating curve creation. See the details of error metrics in Table 1.

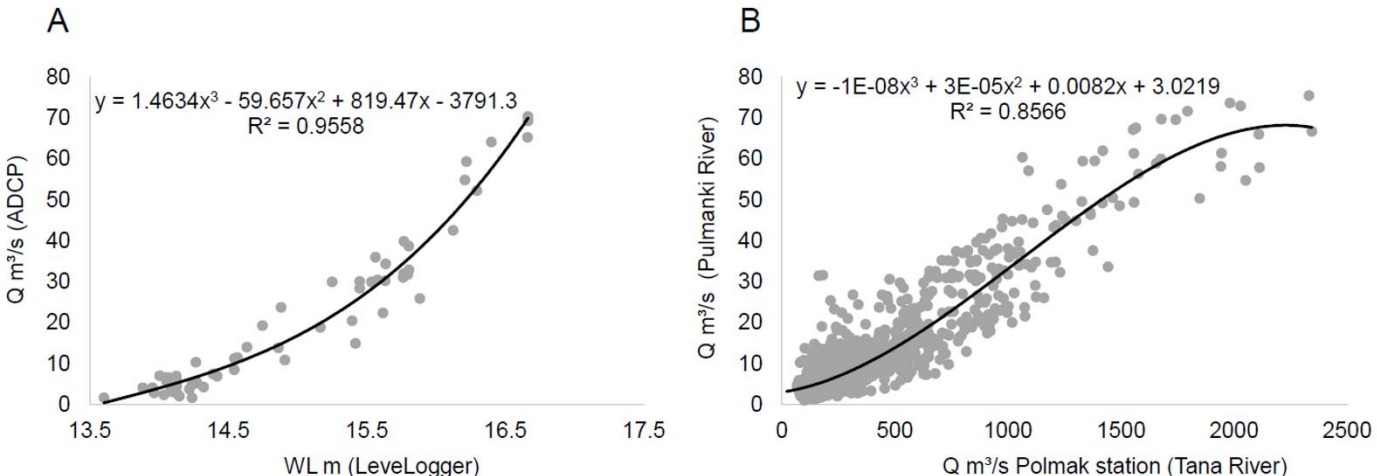

Figure 2. Rating curves for Pulmanki River hydrographs. A) Regression curve of discharge measurements (Q m³/s ADCP) and LeveLogger water level (WL) in Pulmanki River 2008-2023. This polynomial function A was used to calculate hydrographs for years 2008-2023 B) Regression curve showing the relationship between the discharge in Pulmanki ( Q m³/s calculated based on regression curve A) and Polmak (Q m³/s measured, national gauging station) during 2008-2023. This polynomial function B was used for calculating Pulmanki River discharge for years 1992-2007.

Table 1. Error metrics of the final hydrographs derived from two different data sources: LeveLogger discharge data and Polmak Station discharge data. MAE = Mean Absolute Error, SDE = Standard Deviation of Error, r = Correlation Coefficient, n = Number of samples.

| Pulmanki River Q Derived from: | Min. Error (m³/s) | Max. Error (m³/s) | Mean Error (m³/s) | MAE (m³/s) | SDE (m³/s) | r | $R^2$ | n |
|---|---|---|---|---|---|---|---|---|
| LeveLogger | -9.59 | 10.73 | -0.24 | 2.92 | 3.74 | 0.94 | 0.89 | 152 |
| Polmak Station | -51.48 | 20.34 | -0.39 | 2.59 | 4.65 | 0.89 | 0.80 | 1804 |

## 3.2. Hydrograph classification

The hydrographs were classified into distinct flood-event types based on the peak shape in Python program using the SciPy Scientific Python (SciPy) library. A threshold value of 23.46 m³/s (75th percentile, p75 discharge) for flooding was set to classify significant spring flood-events. A sensitivity analysis on peak-finding thresholds was conducted using the 50th, 60th, 70th, 80th, and 90th percentiles. Threshold values at the 70th and 80th percentiles were found to capture the majority of relevant spring flood events, and consequently, the 75th percentile (p75) was selected for this study. The commonly used threshold of the 90th percentile (Q > 58 m³/s in this case) restricted the dataset too severely, with the algorithm failing to detect spring floods in certain years, particularly those with low peak discharges. Furthermore, using the p90 value resulted in hydrographs that included only the very peak of the flood event, without capturing the rising and falling limbs of the hydrograph, which are crucial for evaluating sediment dynamics and flow–sediment interactions. Thresholds below

244 the 70<sup>th</sup> percentile included peaks outside of spring flood season, and thus these thresholds
were not ideal for this study. The definition for high and low flood-event was set to be either
above or below the mean flood discharge of 40 m³/s, respectively.
The event classification was done by estimating different flood peak features such as peak
timing, prominence, peak height, and event duration. First, a Savitzky-Golay smoothing filter
was applied to the dataset to reduce noise and enhance the detectability of flood peaks.
This was accomplished using the Savgol_filter function from the `scipy.signal` module, with
a window size of 11 and a polynomial order of 3 to preserve relevant hydrograph features.
Peak shapes within the smoothed data were identified and classified into distinct flood-
events using the `find_peaks` function from the `scipy.signal` module. The following
parameters and minimum values were found to most effectively identify peak events: the
minimum discharge threshold for a flood event, defined as the 75th percentile (p75 Q), a
minimum hydrograph width of one day, measured from the start of the rising limb to the end
of the recession limb, and a minimum prominence of 2 m³/s, indicating how much the peak
stands out from the surrounding baseline.
Four different event types were detected: A) High one-peak (Q>40 m³/s), B) Low one-peak
(Q<40 m³/s), C) Two separate peaks (Q>p75, Q<p75, Q>p75), and D) Wavy peak (two
Q>p75 peaks) (Figure 3A-D). For modelling purposes, the most typical event of each type
was selected (red solid line in Fig. 3A-D). The precipitation-driven discharge peaks in July,
August and September were left out of the analysis as none of them exceed the flood
threshold discharge of p75. In addition, previous studies indicate that the majority of high-
latitude rivers transport most of their annual sediment load during the main flood event,
namely the spring flood (Syvitski, 2002; Zhang et al., 2022; Blåfield et al., 2024b). Therefore,
the focus of this study was placed solely on spring flood peaks. In this region, spring flood
peaks are driven by climatic factors such as rising temperatures and rainfall, which induce
snowmelt, increase runoff, and lead to the break-up of river ice cover.

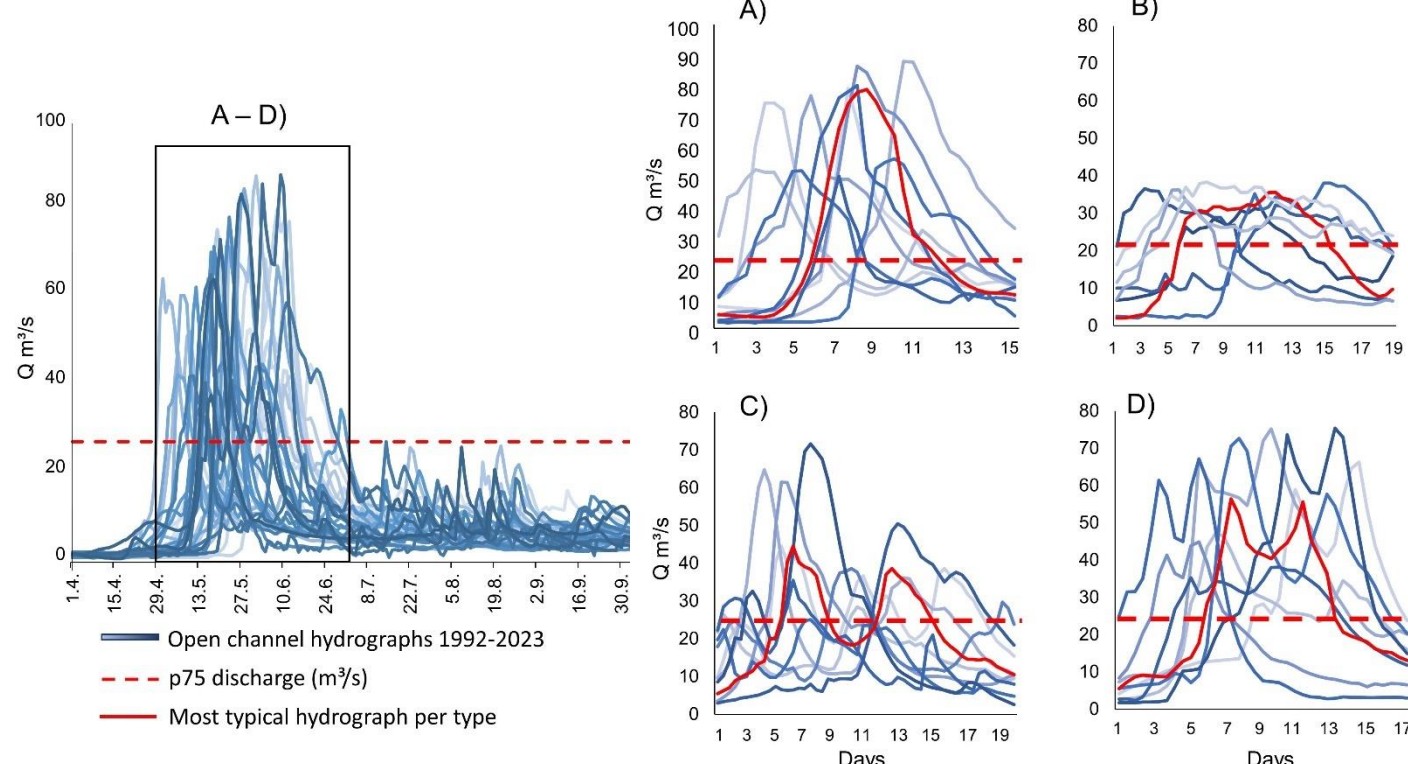

Figure 3. All the generated hydrographs of years 1992-2023. The classification led to four distinct flood-event shapes: A) High one-peak flood, B) Low one-peak flood, C) Flood with two separate peaks, and D) Flood with a wavy peak. The solid red hydrograph is the most typical flood-event of each shape which was thus used in the morphodynamic model. Red dashed line is the 75th percentile threshold discharge for spring flood.

**3.3. Hydroclimatic data and statistical analysis**

Climate data from the Nuorgam weather station (see location in Fig. 1B), 11 metres above the mean sea level and 17 kilometres North from the Pulmanki River study area, was downloaded from the Finnish Meteorological Institutes open data service. Daily Total, Min, Mean and Max temperature, precipitation, and snow depth data of years 1991-2023 were selected for the variance and trend analysis as these variables are closely related to the hydrological properties of rivers (Veijalainen et al., 2010; Irannezhad et al., 2022). Annual Min, Mean, Max and Total values were derived from the daily data and used in the trend analysis (Fig. 4). In addition, duration of snow cover, number of precipitation-days, and occurrence of extreme snow/precipitation events ($95^{th}$ percentile) were derived for the trend analysis. For detailed analysis of springtime trends, the corresponding measures were derived for March, April, and May as well. Only one weather station was included in the analysis as other stations are located 50-100 kilometres away with over 100-meter elevation difference to the area of interest. The year 1991 was included in the climate time-series as the analysis was conducted on hydrological years instead of calendar years.

The Mann-Kendall (M-K) trend test was carried out on all climate variables with $\alpha = 0.05$ significance level to identify statistically significant monotonic trends. In addition to climate variables, the MK-trend test was run on the classified flood hydrographs to examine trends in the occurrence, timing, volume, and duration of each flood-event hydrograph type. Possible serial correlations were removed by using Hamed & Rao (1998) M−K modification which is explained in detail in e.g., Daneshvar Vousoughi et al., (2013) and in Jhajharia et al., (2014). The effect of outliers on the trend was removed by using a non-parametric linear regression Sen's slope estimator (Sen, 1968). Analysis of Variance (ANOVA) with $\alpha = 0.05$ significance level was run to identify possible significant differences between the means of the variables, i.e. whether the annual/cold-season/spring or May weather conditions differ significantly across the four spring flood-event type.

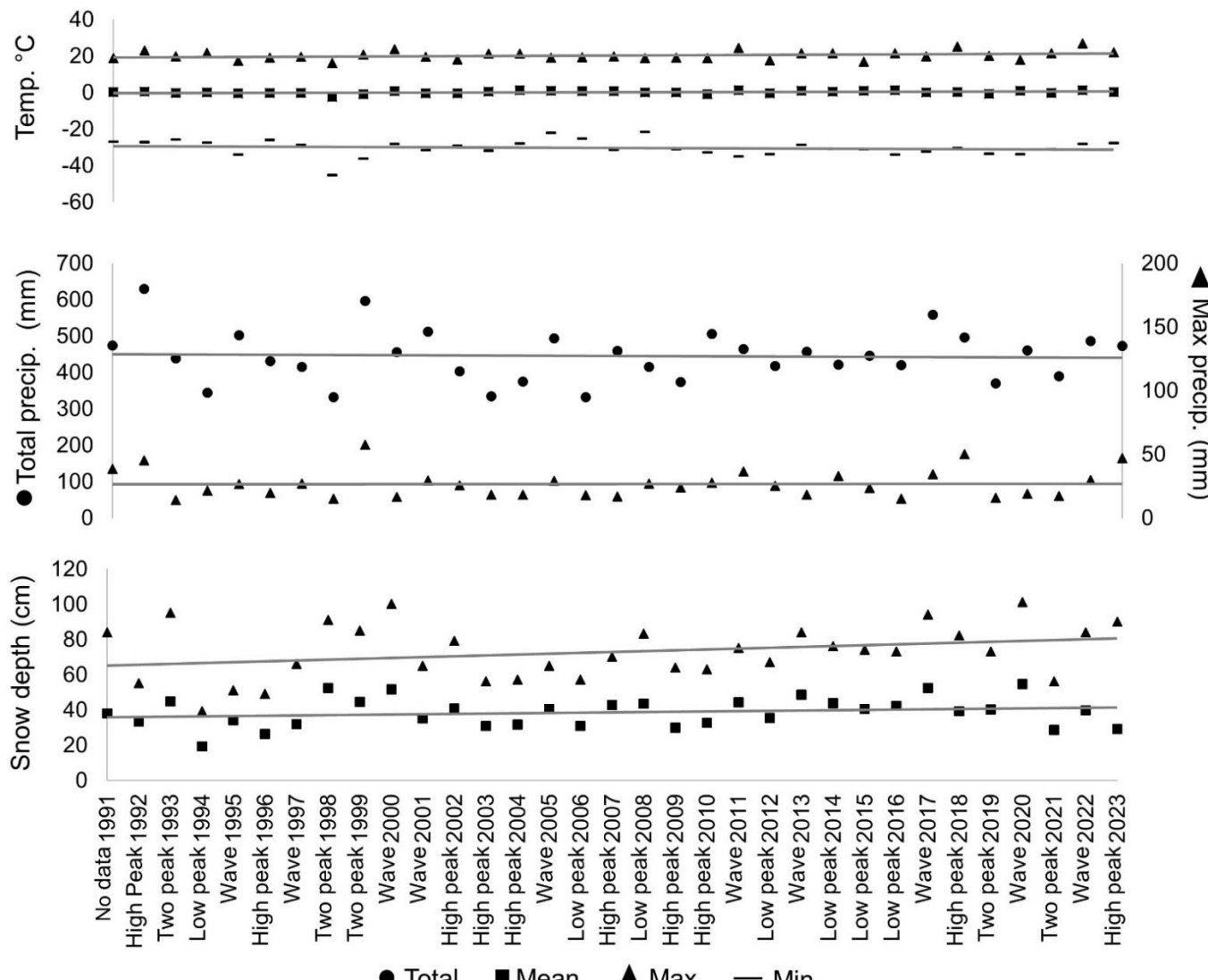

Figure 4. The annual climate time-series of the 32-year time period derived from the daily data. The corresponding flood-event types are marked on the x-axis.

**3.4. Sediment and bedload sampling**

Both grab samples with Van Veen sediment sampler, and bedload samples with Helley-Smith sampler were collected from the riverbed in 2019. A total of 70 grab samples (ca. 500 g) and 24 bedload transport samples were collected during various discharges from the area of interest. Grab samples were collected across the entire 6-kilometre reach during a single autumn field campaign under low discharge (4.2 m³/s) conditions. Samples were taken from the channel bed at left and right bank of each meander inlet, apex and outlet. Bedload transport samples were obtained during both, spring and autumn campaigns, under varying discharge levels (7.5 m³/s, 56 m³/s, and 4.2 m³/s). Twelve bedload transport samples were collected per campaign, each with a sampling duration of six minutes. The samples were dry sieved using half-phi intervals and the amount of material in each sieve was weighted. Sample statistics were calculated in GRADISTAT-program (Blott & Pye, 2010) using the Method of Moments which is based on a logarithmic distribution of sample phi sizes. GRADISTAT utilises its own scale with only four classes (Silt, 0.002–0.063 mm, Sand,

0.063–2 mm, Gravel, 2–64 mm and Boulders 64–2048 mm). The results of sediment and
bedload sampling were utilised in the morphodynamic model as multiple sediment fractions,
spatially varying Manning's Roughness parameter, and for calibrating and validating the
sediment transport rates (see details in Blåfield et al., 2024b).

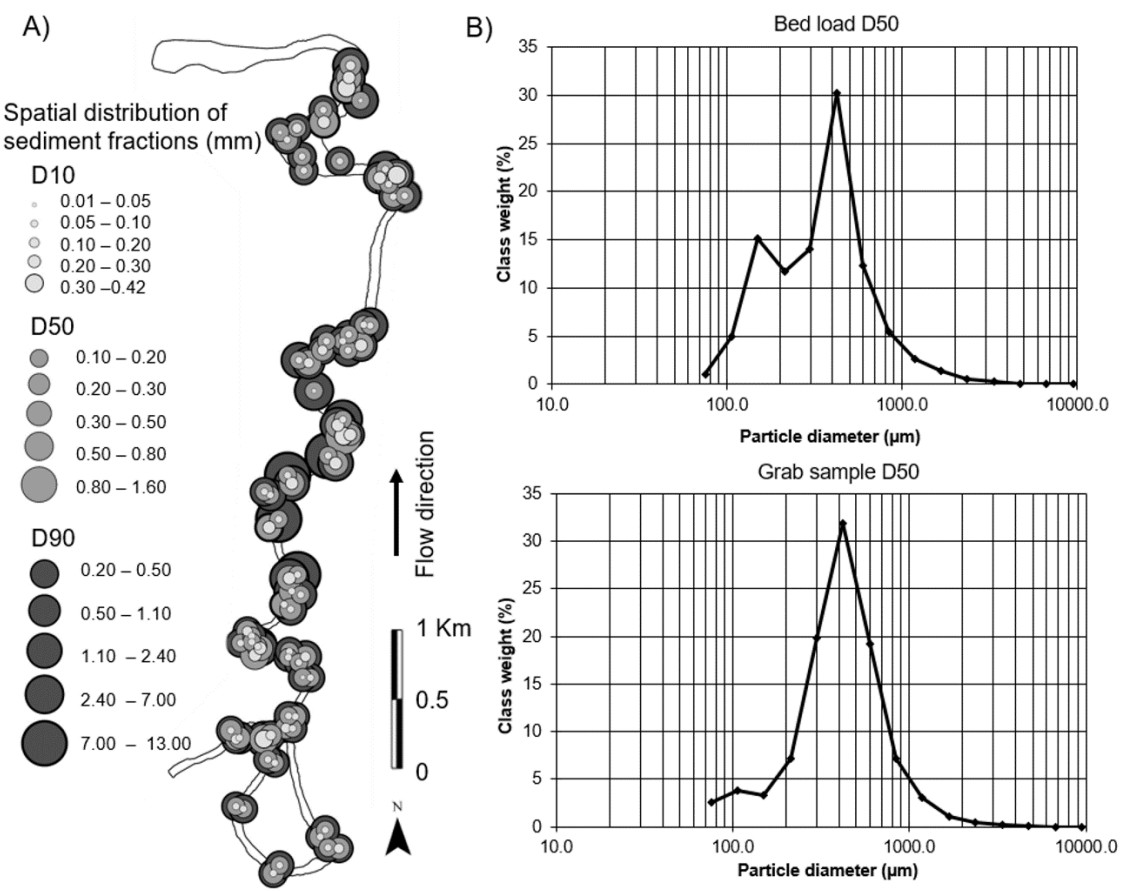

Figure 5. A) Spatial distribution of sediment fractions D10, D50 and D90 based on the
collected field samples. B) D50 particle diameter distribution of all the collected bedload and
grab samples in micrometres.

## 3.4 Morphodynamic modelling

The authors have previously presented and validated the model used in this study (Blåfield
et al., 2024b). In this study, four distinct flood-event hydrographs (A-D in Table 2.) were
simulated using the same initial channel geometry and sediment composition. A depth-
averaged morphodynamic model with curvilinear, unstructured grid of 2x2 meter cell size
was built utilizing FLOW 2D-module of Delft3D software. The model geometry was based on
a digital elevation model derived from Structure-from-Motion (SfM). Specific details of the
SfM creation can be found in Blåfield et al., (2024b), and general from Micheletti et al.,
(2013), and Dietrich et al., (2017). Multiple sediment fractions and spatially varying
Manning's Roughness based on the grab sediment samples from the field were used, as
these additions have been shown to significantly improve predicted morphodynamics (Kasvi
et al., 2014). Each simulation featured hourly varying discharge conditions to evaluate
sediment transport dynamics, sediment transport hysteresis patterns, and morphological
responses to the shape and sequencing of the simulated hydrographs. The model time-step

was set to 0.05 minutes, with both spin-up and output intervals set at 720 minutes. Morphology, source and sink terms, and total sediment transport were updated at each time step. The model solved morphology independently based on the source and sink terms of van Rijn (1993) approach. Transport boundary conditions, i.e., sediment feeding into the model, were solved using the Neumann law and updated at each time-step. This allowed the model to dynamically adjust the sediment supply and concentration at the inflow to match the internal model conditions, thereby minimising accretion near the model boundaries. Subsequently, sediment transport hysteresis and geomorphic activity for each flood-event type were calculated from the source and sink terms, as well as from the modelled total volume of sediment mobilised within the inundated area. The default scheme for dry-cell erosion of banks was applied without further adjustment, as the focus of the study was on longitudinal sediment transport and vertical changes to the channel bed. Detailed parametrisation, as well as the model's calibration and validation are provided in Blåfield et al., (2024b).

Delft3D is unable to simulate ice-covered flows or the effects of freeze–thaw processes on bank erosion. These limitations, together with the absence of vertical flow representation in the two-dimensional simulation, introduce simplifications into the modelling of flow dynamics and sediment transport. The use of user-defined parameters further contributes to uncertainty, particularly in the spatial and temporal patterns of erosion, transport, and deposition. The van Rijn (1993) approach is sensitive to user-defined parameters such as sediment fraction, composition, and associated threshold conditions (Pinto et al., 2006). However, Kasvi et al. (2014) demonstrated that the van Rijn formulation performs more reliably when applied with spatially variable, field-based sediment fractions and Manning's roughness coefficients rather than uniform values. While the van Rijn transport formula typically produces lower transport rates than other formulations (Schuurman et al., 2013; Kasvi et al., 2014), it remains widely regarded as the most physically based and reliable method (Pinto et al., 2006; Kasvi et al., 2014). The user-defined parametrisation used in the present study is detailed in Blåfield et al. (2024b). Spatial variability in sediment grain size and Manning's roughness, alongside the inclusion of medium transverse bed slope effects, were identified as key parameters influencing sediment load predictions and morphological change (Nicholas, 2013; Kasvi et al., 2014), and were prioritised for refinement during simulation.

Table 2. The details of each model run. The flow conditions of flood-events A-D are based on the hydrograph classification in section 3.2. The morphological parameters are based on the sediment and bedload sampling from the field.

| Event | Duration (days) | Peak Q m³/s | Total Q Volume m³ | Sediment Supply | Morphology | Sediment composition |
|---|---|---|---|---|---|---|
| A | 7 | 80 | 29 868 586 | Feeding | Sand bed | Sand, Gravel |
| B | 13 | 35 | 34 851 505 | Feeding | Sand bed | Sand, Gravel |
| C | 14 | 48 | 26 2383 45 | Feeding | Sand bed | Sand, Gravel |
| D | 9 | 60 | 31 20 1609 | Feeding | Sand bed | Sand, Gravel |

## 4. Results

### 4.1 Hydroclimatic conditions and flood-event type variability

The variance analysis of flood events of types A–D and the prevailing climatic conditions indicated that the climatic conditions of the preceding hydrological year were the most significant of the tested variables influencing the type of spring flood event (Table 3). Other significant factors influencing the flood-event type included the cold season (October–May) mean temperature, spring rainfall (March–May), and May warmth, expressed as the cumulative temperature sum in May (Table 3). In addition to climatic conditions, the timing of peak discharge varied significantly between event types. The number of snow cover days in May demonstrated a trend approaching statistical significance (Table 3). By contrast, rainfall during the cold season, May rainfall, and the spring mean temperature did not exhibit significant differences between flood-event types (Table 3).

Flood events of Type A were typically associated with high annual snow accumulation, low annual temperatures, and rapid warming in May, resulting in a low number of snow cover days during May (Fig. 6). These events also experienced high annual precipitation but low spring rainfall (Fig. 6). Thus, flood events of Type A can be characterised as occurring in cold, snow-rich years, where rapid warming in May leads to sharp and high flood hydrographs. Flood events of Type B were associated with the warmest cold season mean temperatures, along with moderate spring rainfall, snow accumulation, and cumulative May temperatures (Fig. 6). These conditions suggest that snowmelt may begin during the cold season and continue through spring, resulting in reduced energy availability during the main melt period in May. Flood events of Type C were linked to the lowest annual precipitation, the lowest cold season temperatures, and the coldest May temperatures (Fig. 6). However, these events also exhibited the highest snow accumulation during spring. Overall, Type C floods reflect dry, mixed, or transitional climatic conditions, in which a particularly cold winter and spring lead to delayed snowmelt. This delayed melt, when combined with May rainfall and variable temperatures, may result in two distinct melt peaks. Flood events of Type D were characterised by high annual and spring snow accumulation, alongside the warmest annual and cold season mean temperatures, but relatively low temperatures in May, leading to a prolonged persistence of snow cover during May. These events also experienced high annual precipitation and considerable variation in spring rainfall. Consequently, this flood type typically occurs following a warm and wet year, when May is cold and experiences highly variable rainfall, resulting in non-uniform snowmelt and the development of wavy hydrographs.

Table 3. Results of one-way ANOVA test on the main variables with α = 0.05 significance level. Statistically significant p-values are bolded. T = Temperature, P = Precipitation, Cold season = October-May, Spring = March, April, May.

| Variable | F-statistics | p-Value |
|---|---|---|
| Annual mean T | 3.73 | **0.022** |

| | | |
|---|---|---|
| Annual snow sum | 7.73 | **0.006** |
| Annual total P | 4.00 | **0.017** |
| Cold season mean T | 3.38 | **0.032** |
| Cold season Rainfall | 2.26 | 0.104 |
| Spring mean T | 1.78 | 0.174 |
| Spring snow sum | 1.93 | 0.103 |
| Spring rainfall | 3.13 | **0.050** |
| May cumulative T | 3.41 | **0.032** |
| May n. of snow cover days | 2.36 | 0.083 |
| May Rainfall | 0.97 | 0.420 |
| Peak Q timing | 3.28 | **0.035** |

420

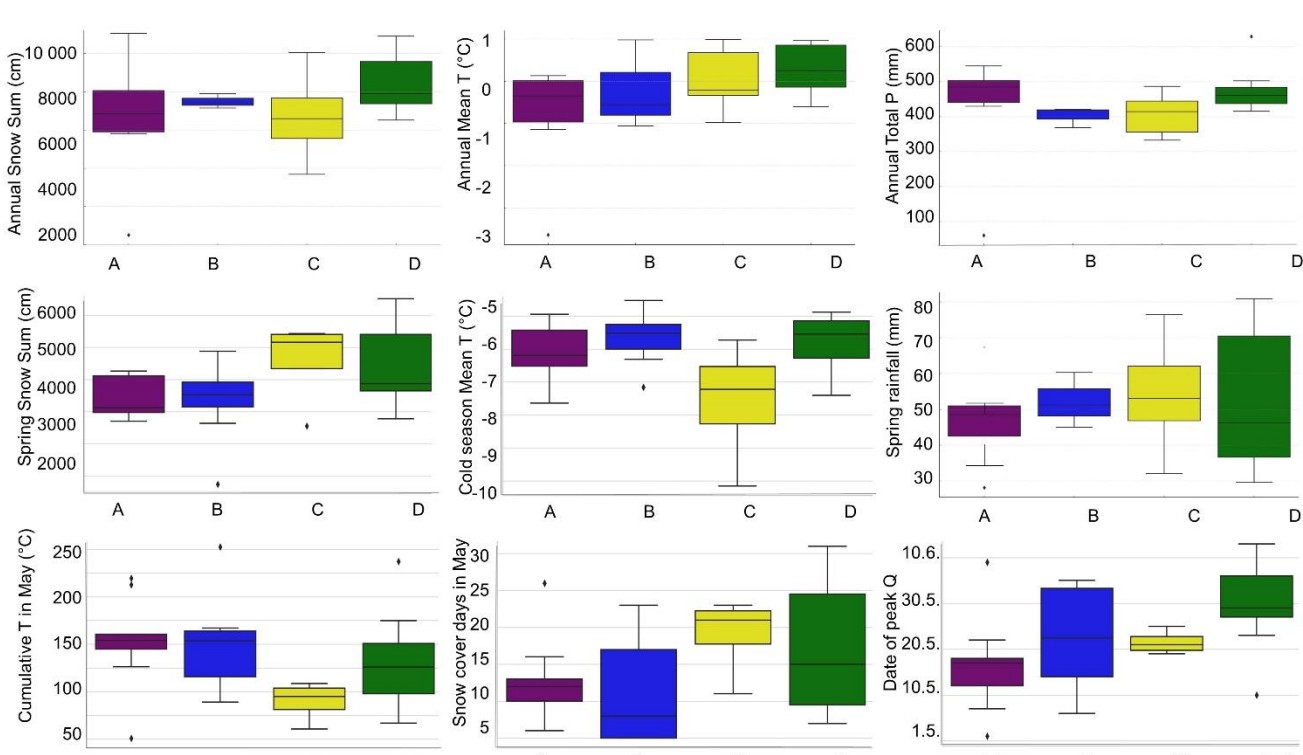

421

Figure 6. Statistically significant differences between the variable means illustrated in the boxplots, showing the distribution, median, and variation of climate variables associated with each flood-event type. Two borderline variables close to significance (May snow cover days and spring snow sum) were plotted as well.

The wavy (D) and high one-peak (A) events appeared the most frequently, both occurring 10 times within the 32-year time-series. The wavy events of type D had an average duration of 9 days whereas high one-peak event of type A lasted 7 days on average. Low one-peak events of type B occurred 7 times and had the longest average duration of 13 days. Finally, the two-peak events of type C were the least frequent type with only five occurrences lasting 14 days on average. No significant trends were observed in re-occurrence, duration, volume, or timing of any of the flood-event types within the 32-year time-series (Fig. 7). Trend analysis on the climate variables indicate that in snow-related variables (mean, maximum, and extreme snow), all annual trends (square marker) were positive with statistically significant

increase. The max snow amount had statistically significant trend also in spring (March-May, circle marker). The number of snow days, however, showed a non-significant weakly negative trend (Fig. 7). Temperature trends were mostly positive, with statistically significant increases in both annual and spring mean temperature (Fig. 7). Spring-time max temperature had significant increasing trend indicating that especially springs have gotten warmer over the time-series. Minimum temperature showed non-significant trend in annual and spring-time data. Precipitation-related trends were more variable. Mean and max precipitation exhibited mostly negative non-significant trends or no trend at all, while the annual extreme precipitation (95th percentile) showed significant decreasing trend (Fig. 7).Even though the spring-time precipitation did not indicate significant trends in volume, the number of precipitation days had significant increasing trend.

Overall, the results suggest that while the frequency and characteristics of individual flood-event types have remained relatively stable over the 32-year period, underlying climatic drivers have undergone notable changes. In particular, the increase in snow accumulation and rising spring temperatures point toward a shift in the timing and dynamics of snowmelt, even if not yet reflected in observable flood trends. The significant rise in the number of precipitation days during spring, despite no clear trend in total precipitation volume, may also contribute to more fragmented or prolonged runoff events, potentially supporting the occurrence of events of type B and D.

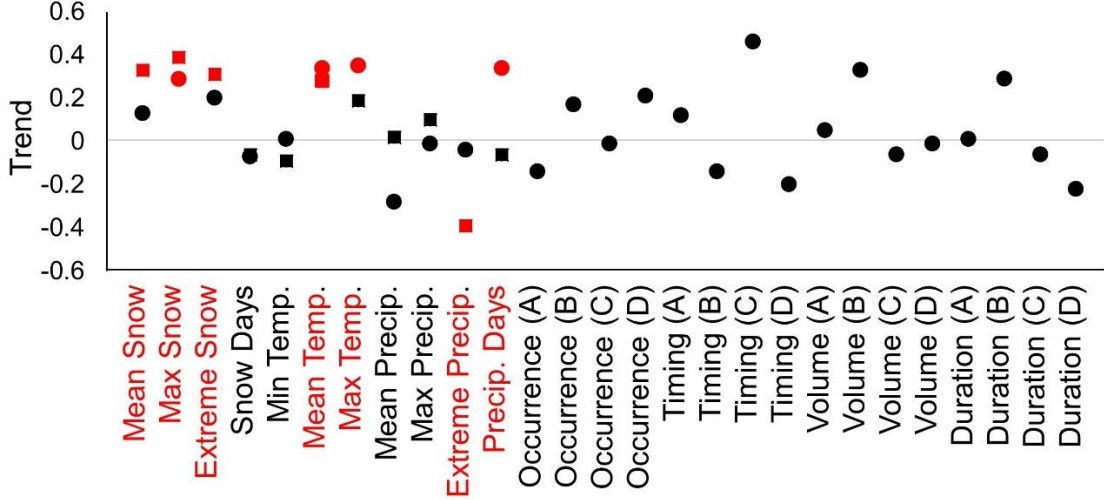

Figure 7. The M-K-trend test results of the climate-related variables during the 32-year study period. Red markers indicate statistically significant trends and black markers non-significant. Square markers represented annual trends, while circles represent seasonal trends in spring (March-May).

**4.2 Morphological response to sediment transport hysteresis**

The modelled results suggest that hydrograph shape may have a significant influence on morphological response and sediment transport hysteresis. Both the total transported sediment (TTS, calculated across the entire model area) and the type of sediment transport hysteresis appeared to vary across the modelled events. The wavy event (Type D) was associated with the largest volume of TTS, with the first peak contributing approximately 59% and the second peak 41% of the event's TTS. Thus, the transport rate during the first peak was about 28% higher than during the second peak. In the flood event characterised

by two separate peaks (Type C), the first peak contributed 63% and the second 37% of the total TTS. Consequently, the transport rate during the second peak was approximately 42% lower than during the first. The TTS of event C was around 17% lower than that of event D. The high one-peak event (Type A) yielded a TTS volume approximately 4% lower than event D and about 11% higher than that of event C. In contrast, the low one-peak event (Type B) exhibited a TTS volume about 30% lower than that of the high one-peak event (Type A), and approximately 20–32% lower than the double-peaking events C and D, respectively.

All events predominantly exhibited counterclockwise sediment transport hysteresis, where the transport peak occurred after the peak discharge (Fig. 8A–D), suggesting that sediment transport lagged behind changes in discharge and flow conditions. However, the modelled sediment transport hysteresis loops appeared to vary in complexity and shape depending on the flood-event type. The single-peak events (Types A and B) displayed relatively simple counterclockwise loop-shaped hysteresis, with sediment transport following the peak discharge (Fig. 8A–B). Event C appeared to exhibit a more complex hysteresis pattern, including multiple counterclockwise loops, which may indicate that sediment mobilised during the first peak was partially deposited between the peaks, as the second peak showed significantly lower TTS (Fig. 8C). In the wavy event (Type D), the first peak exhibited counterclockwise hysteresis, whereas sediment transport during the second peak appeared to precede the second discharge peak, resulting in clockwise hysteresis (Fig. 8D). This complexity may reflect variability in sediment mobilisation processes and sediment availability. Across all events, higher TTS values were observed during the falling limb compared to the rising limb at corresponding discharge values, suggesting that sediment transport was not directly proportional to discharge (Fig. 8A–D). This discrepancy highlights the potential influence of delayed and progressive sediment mobilisation, as well as the lagged morphological response of bedforms. These findings imply that flood-event shape may have a considerable influence on sediment transport hysteresis and, consequently, on riverbed morphological development.

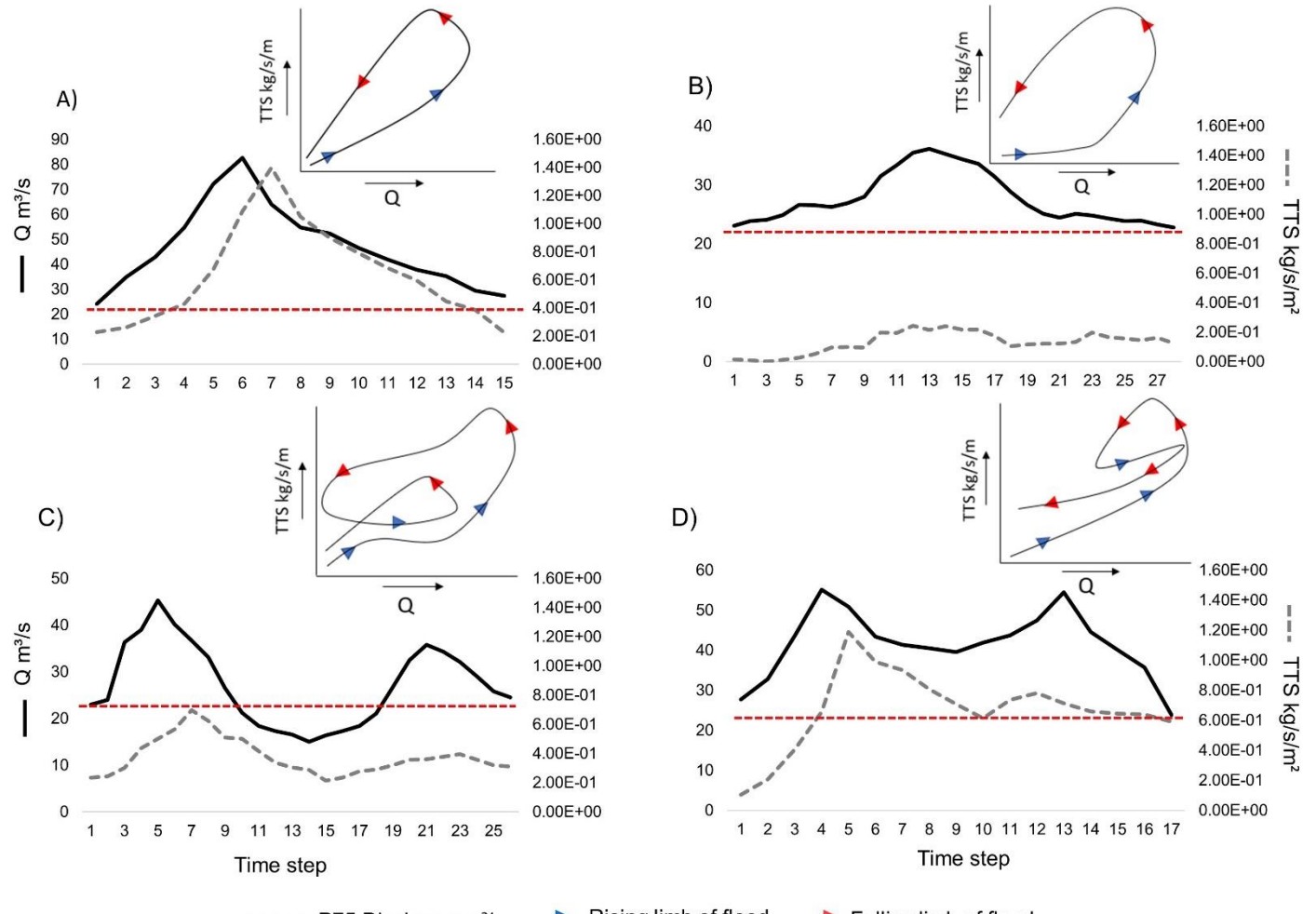

Figure 8. The modelled flood-event hydrographs and sediment load at each timestep. On the upper right corner of each graph is the sediment transport hysteresis of the event type. The blue arrows indicate rising limb and red arrows falling limb of the flood. The red dashed line shows the threshold p90 discharge. A) High one-peak event and sharp counterclockwise sediment transport hysteresis. B) Low one-peak event and wide counterclockwise sediment transport hysteresis. C) Event with two separate peaks and counterclockwise sediment transport hysteresis with a loop. D) Wavy type event and hysteresis loop with counterclockwise and clockwise directions.

Each modelled event appeared to demonstrate different patterns of morphological response (Fig. 9A–D), influenced by variations in sediment transport hysteresis, stream power, and flow velocity. Event A produced the second highest total volume of mobilised sediment and geomorphic activity (Fig. 9A). Based on the model, this event appeared to experience the most extensive erosion throughout the reach, with deposition areas remaining relatively localised. The highest stream power values were modelled in this event, exceeding 24 W/m², alongside a mean flow velocity of 0.61 m/s, both of which likely contributed to substantial erosion and an overall net sediment loss of –14,772 m³. Sediment input from upstream was insufficient to compensate for this loss. In contrast, event B exhibited the lowest geomorphic activity, with a more balanced distribution of erosion and deposition across the river reach, resembling classical meander behaviour with distinct riffles and pools (Fig. 9B). Stream power during this event was considerably lower, predominantly below 10 W/m², with a mean

flow velocity of 0.36 m/s. These conditions likely facilitated the deposition of eroded and transported sediment within the reach, resulting in a net sediment gain of 5,482 m³.

Event C showed a relatively balanced response, with an even distribution of erosion and deposition, and the smallest net change, resulting in a sediment gain of 1,132 m³ (Fig. 9C). The upstream section experienced the greatest erosion, while sediment accumulation was most pronounced downstream. Only minor changes occurred in the middle reach based on the model. Stream power for event C was moderate, with values mostly below 16 W/m² and only occasional exceedances above 20 W/m². Event D exhibited the most fragmented morphological response, with small, scattered areas of both erosion and deposition distributed throughout the reach (Fig. 9D). The stream power distribution for event D was more similar to that of event A, with values exceeding 20 W/m² and a mean flow velocity of 0.54 m/s. Despite the relatively high energy, event D produced a more balanced sediment budget, though it still resulted in a net sediment loss of –6,267 m³. Geomorphic activity per unit area appeared highest for events A and D, both of which showed considerable sediment mobilisation but resulted in different morphological responses likely due to hydrograph shape. Events B and C exhibited lower geomorphic activity, with a tendency towards sediment deposition rather than erosion. The pattern of morphological change associated with the modelled flood events thus appeared to be linked to the peak shape, sequencing, and the resulting sediment transport hysteresis patterns, which collectively influenced the morphological response of bedforms.

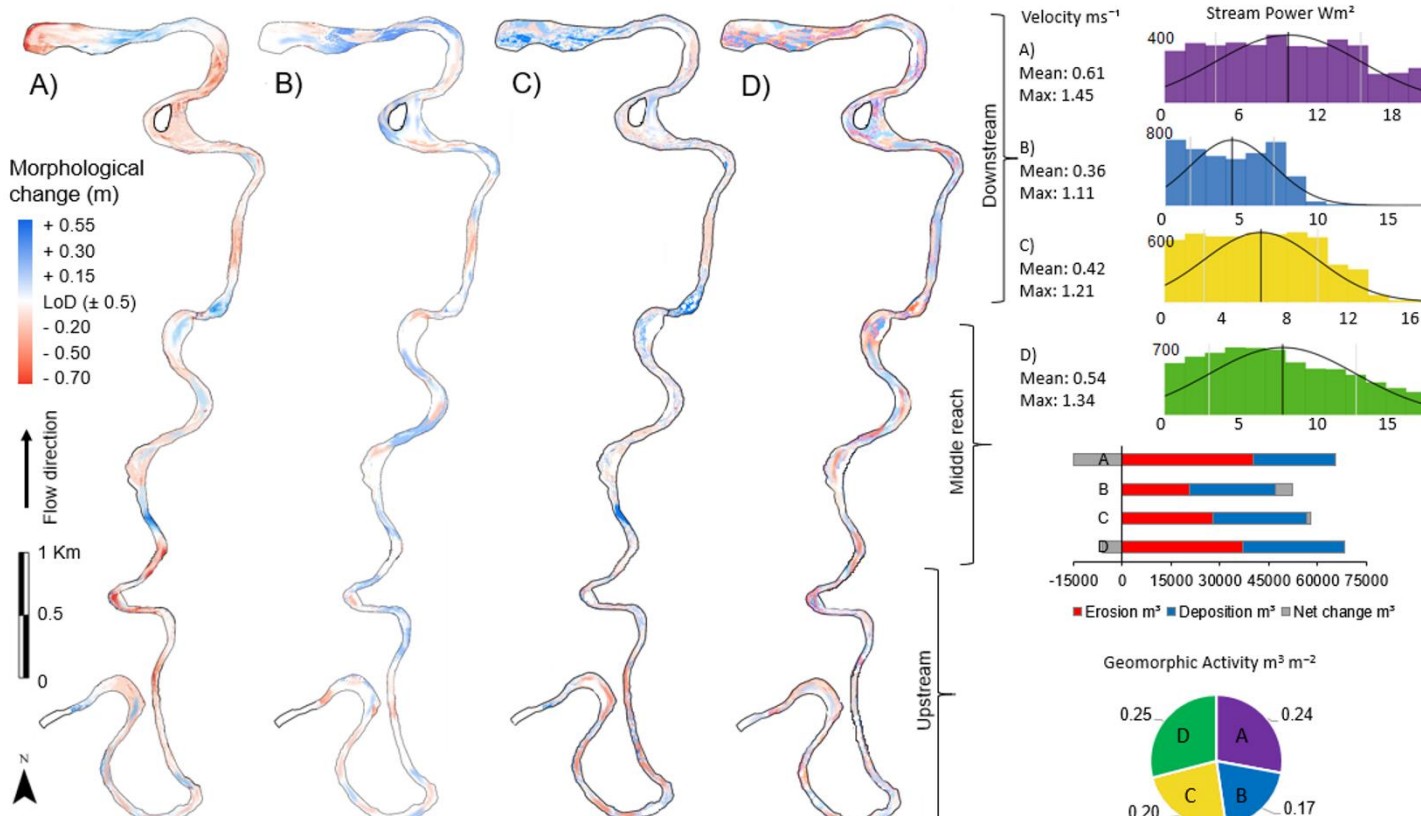

Figure 9. Morphological adjustment of each flood-event (A-D) in left panel: A) Distinct areas of heavy erosion and deposition. B) Desecrate morphological changes but distinct areas of erosion and deposition. C) More complex morphological changes patched around the river reach. D) Heavy erosion and deposition spread complexly inside the reach. Right panel: A-

D events mean and max velocity, histograms of stream power (x) distribution within number of model cells (y), volume of erosion, deposition, and net change, and geomorphic activity.

## 5. Discussion

### 5.1. Flood-event types and hydroclimatic conditions

The results of variance analysis and trend test of climate variables and flood-event types aligned with well-documented responses to climate change in cold regions (Cockburn & Lamoureux, 2008; Vormoor et al., 2016; Matti et al., 2017; Arp et al., 2020). The significant increase in both mean and maximum spring temperatures matched global climate model predictions for continued warming at high-latitudes (Koenigk & Brodeau, 2017; Huo et al., 2022). The increased snow depth also aligned with Pulliainen et al. (2020), who reported rising snow accumulation and snow water equivalent (SWE) in the studied region. Despite this, no significant changes in flood volumes were observed, consistent with previous studies in Fennoscandia (Veijalainen et al., 2010; Korhonen & Kuusisto, 2010; Matti et al., 2017; Lintunen et al., 2024). This lack of change was attributed to milder winter conditions and longer snowmelt period resulting from warming temperatures, which lead to more stable runoff during spring (Fischer & Schumann, 2019; Zhang et al., 2023). Additionally, no significant trends were found in the timing, duration, or interval of flood-events, consistent with earlier research in snowmelt-dominated regions (Veijalainen et al., 2010; Vormoor et al., 2016; Matti et al., 2017).

Despite the absence of significant trends, low-peak floods (B) increased in both volume and duration, while wavy floods (D) showed a reduction, respectively (Figure 7). Based on the results of ANOVA, both flood-events of this type were influenced by high annual temperature and high snow accumulation, but significantly different spring-time and May weather conditions. Events of type B experienced long and warm melt period during spring whereas events of type D were associated with late spring warmth with varying amounts of rainfall leading to non-uniform runoff. The climatic conditions associated with these event types are expected to intensify across the Northern Hemisphere (Callaghan et al., 2012; Kunkel et al., 2016; Conolly et al., 2019; Pulliainen et al., 2020; Hu et al., 2023), although climate change impact on snow accumulation is likely to vary spatially. These event types also exhibited an increase of re-occurrence indicating that these flood-event types are likely to become more common in the future. High one-peak floods (A), however, were associated with cold, snow-rich years, where rapid warming in May leads to sharp and high flood hydrographs. This is consistent with findings that cold springs delay snowmelt and ground thaw, leading to high discharge peaks when the thaw eventually occurs (Labuhn et al., 2018). Unlike double-peaking floods, single-peak events involved lower temperatures and rainfall during spring, and therefore the rain-on-snow effect could be linked to the wetter conditions typical to events of type D. Even though events of type C are also double peaking, these hydrographs were linked to dry, mixed, or transitional climatic conditions, in which a particularly cold winter and spring lead to delayed snowmelt. Hydrographs of event type C had however, significant

amount of rainfall in May which together with cold temperature conditions likely causes the two separate melt peaks typical to this event.

Climate change is expected to increase annual temperatures and to modify the precipitation patterns in high-latitude areas (Zhang et al., 2023; Blöschl et al., 2017). These changes will likely have an impact on the occurrence of certain flood-event types. Increased spring rainfall can increase rain-on-snow events significantly amplify runoff and flood peaks, particularly together with deep snow packs and accelerated snowmelt from warmer spring temperatures. Similar pattern have been recognized previously on high-latitudes by Fischer & Schumann (2019). The results observed in this study point to the direction of possible future hydroclimatic regime shift. These findings highlight the complex effects of climate change on flood-events and underscore the importance of considering flood-event sequencing in assessing the impacts of hydroclimatic shifts. Future research could explore climate teleconnections, such as the North Atlantic Oscillation (NAO) or Arctic Oscillation (AO), to better understand the conditions driving specific flood-events (Dahlke et a., 2012; Villarini et al., 2013; Irannezhad et al., 2022). In addition, it is worth of noting that while interpreting the results of this study, especially the results of ANOVA, the sample size has a critical impact on the reliability and validity of the results. Larger sample sizes increase the statistical power of the test, improving the ability to detect true differences between group means (Lakens, 2022). They also provide more precise estimates of means and variances, reduce the influence of outliers, and help satisfy the assumptions of normality and homogeneity of variances. In contrast, small samples can result in underpowered tests, unstable F-statistics, and greater sensitivity to assumption violations, ultimately reducing the robustness of the findings

## 5.2. Flood-event types and morphological response

When interpreting the morphological results of the simulations in this study some limitations should be considered. As a depth-averaged model, it did not resolve vertical flow structures or secondary currents, which limit the capacity to fully represent sediment transport and bank erosion processes (Pinto et al., 2006;Nicholas et al., 2014; Williams et al., 2014). In addition, the model lacks the ability to simulate ice-covered flows and freeze–thaw effects, both of which significantly influence sediment dynamics and channel stability in cold-region rivers (Zhang et al., 2022). Model sensitivity to user-defined parameters, such as sediment fractions and roughness coefficients, further contributes to output uncertainty (Pinto et al., 2006). Moreover, the use of morphological acceleration factors and simplified boundary conditions may exaggerate or underrepresent morphological processes. Consequently, while the simulation runs presented in this study are effective in assessing relative differences between scenarios, caution is necessary when interpreting absolute sediment budgets and localised morphological changes due to simplifications, and the fact that it cannot simulate ice and freeze-thaw effect on sediment transport and bank erosion. In addition, only one hydrograph was modelled for each flood-event type, meaning that selecting a different hydrograph of the same type could have resulted in different volumetric changes and total amounts of transported sediment, as these depend on the flood's volume and transport capacity. However, the findings of this study, together with previous laboratory (Mao, 2012; 2018) and modelling studies (Kasvi et al., 2015), support the view that the shape

and sequence of the hydrograph play a crucial role in determining morphological outcomes. Therefore, it is likely that selecting different hydrographs for each flood-event type would have produced similar types of morphological patterns relative to the flood's magnitude and transport capacity, even though the quantitative results may have differed.

Nevertheless, the model simulations showed that the channel bed's morphological response was influenced by flood-event type and sequences, as well as sediment transport hysteresis pattern, rather than just flood magnitude. Similar finding have been made by Kasvi (2015) who found that the flood duration and flow characteristics have notable impact on channel morphology. All events exhibited dominant counterclockwise hysteresis, common in sand-bed rivers with upstream sediment supply and bedload-dominated transport (Tananev, 2015; Gunsolus & Binns, 2017). However, the riverbed's morphological response varied depending on the modelled hydrograph shape. Single-peak events (A and B) produced distinct erosion and deposition patterns, while double-peaking events (C and D) led to fragmented, small-scale morphological features. Particularly, event B formed classic riffles and pools, typical to meandering rivers (Hooke, 2003, Salmela et al., 2022), whereas the reduced sediment transport during second peaks in double-peaking events, also noted in previous flume experiments (Martin & Jerolmack, 2013; Mao, 2018), resulted in more complex, small-scale bedforms.

The reduction in sediment transport during the second flood peak has previously been linked to bed surface reorganisation, notably the exposure of coarser material (armouring) and infiltration of finer sediments (kinetic sieving), both of which stabilise the bed and increase the energy required for sediment remobilisation (Curran & Waters, 2014; Dudill et al., 2017; Ferdowsi et al., 2017; Mao, 2018). In contrast, event D displayed clockwise hysteresis during the second peak, suggesting that the bed did not fully stabilise between peaks, enabling more rapid sediment remobilisation and resulting in higher total transported sediment (TTS) compared to other flood events. This pattern may also indicate increased input of finer sediments from bank erosion, which tends to intensify during the falling limb of the hydrograph as water levels decline (Lotsari et al., 2014; Lotsari et al., 2024; Yang et al., 2024). Bank erosion dynamics are further influenced by freeze-thaw processes and the presence of seasonally frozen ground, which the current model does not represent. The thermal condition of banks and bars during different flood stages strongly affects erosion rates, with thawed banks being considerably more erodible than frozen ones (Lotsari et al., 2024; van Rooijen & Lotsari, 2024; Yang et al., 2024). Moreover, both soil moisture and the number of freeze-thaw cycles reduce bank stability (Li et al., 2022; Lotsari et al., 2024). As climate change is expected to prolong freeze-thaw periods (Blåfield et al., 2024a; Sha et al., 2025), future bank erosion rates and sediment fluxes are likely to increase. Additionally, enhanced cold-season discharge and earlier freshet onset under warmer conditions will further promote riverbank erosion in many regions (Brown et al., 2020).

The fragmented bedforms from double-peaking floods were likely caused by secondary bedforms cannibalizing the larger topography from the first peak, a phenomenon observed in flume studies (Wilbers & Brinke, 2003; Martin et al., 2013). In addition to flood hydrograph

shape and hysteresis pattern, sediment particle size played a key role in morphological adjustment. The middle reach with the largest particles (Fig. 5) was eroded mainly during events A and D, while events B and C caused minimal change in this section of the river. This finding was consistent with earlier research on particle size impact on sediment transport hysteresis and remobilisation of the sediment particles (Mao, 2012; Malutta et al., 2020).Despite variations in the modelled runoff volumes, the study identified distinct morphological response patterns for each flood-event type. These patterns, shaped by sediment transport hysteresis, distribution of sediment particle size and flood-event sequences, align with findings from previous studies (Martin & Jerolmack, 2013; Gunsolus & Binns, 2017; Mao, 2018). The results highlighted the crucial role of different flood-event types in shaping river morphology, revealing that, while event variation likely helps maintain channel equilibrium in long-term, prolonged exposure to certain events—such as high-energy or multi-peaking floods—could disrupt this balance. Such evolution have the potential to destabilise the channel, by altering sediment connectivity, transport processes, and ultimately the morphological structure of the river systems (Bracken et al., 2015; Zhang et al., 2023). Understanding these responses is essential for predicting future river behaviour and managing morphological stability.

## 5.3. Forecasted hydroclimatic shift and long-term morphological adjustment

This study highlighted the importance of understanding how fluvial sand and gravel-bed systems respond to climatic conditions, particularly by examining the sequences of flood hydrographs, which are often overlooked, and more focus is paid on factors like flood volume, timing, or frequency. The results revealed that flood-event type and peak sequencing had significant impact on the morphological response of the channel. This together with the observed trends, suggested that even in regions, like the one studied, where hydroclimatic changes are not yet fully visible (Veijalainen et al., 2010; Lintunen et al., 2024), flood-event characteristics are evolving with consequences to the river morphology. This and the overserved trends in the hydroclimatic variables underscores that hydroclimatic change is not uniform in space and time across cold regions and rivers should be assessed at the catchment scale to predict future morphological adjustment accurately.

The increase (decrease) of double (single) peaking floods could lead to changes in river system stability, sediment loads, and in the spatial distribution of long-term morphological adjustment if certain type of morphological response begin to accumulate (Bracken et al., 2015; Zhang et al., 2023; Blåfield et al., 2024a). Furthermore, previous research findings suggesting that sediment loads in cold regions could rise by 20-30 % for every 1-2 °C increase in temperature (Syvitski et al., 2002; Li et al., 2021) was supported by this study, as the double-peaking floods related to warmer annual temperatures, showed higher geomorphic activity and sediment loads compared to single peaking events of similar volume. The temperature increase together with altered morphological response pattern could eventually lead to sediment transport regime shift. However, the anticipated shift is likely to be a gradual process (Zhang et al., 2023), and the river system may eventually stabilise again. Yet, before stabilizing the shift is likely to challenge the river channel stability,

making the long-term morphological adjustment, like meander migration, less predictable (Wohl et al., 2017; Hopwood et al., 2021).

Shifts in the sediment transport regime, along with changes in morphological response and long-term adjustment to evolving flood patterns, are likely to influence the morphological response to summer and autumn precipitation by altering sediment availability and bed form composition. Although these precipitation peaks were not the focus of this study, these seasonal peaks should be considered when predicting and evaluating long-term morphological adjustment of river channels as the distribution of seasonal sediment load is likely shifting towards summer and autumn peaks (Li et al., 2021; Zhang et al., 2023; Blåfield et al., 2024a). This could have significant implications for river ecosystems, flood risk management, and infrastructure planning (Beel., et al., 2021; Gupta et al., 2021; Najafi et al., 2021). As discharge regimes become increasingly event-driven rather than seasonally predictable, traditional models of sediment flux that assume clear seasonal patterns may no longer be applicable. Hysteresis patterns, where sediment concentration and water discharge are no longer linearly related, can reveal critical thresholds, sediment sources, and system memory that are key to predicting future river behaviour. Therefore, future research should focus on understanding the combined effects of flood-event sequencing, changing precipitation patterns, and sediment transport dynamics under evolving climatic conditions. Long-term monitoring and advanced modelling efforts will be essential to predict the future morphological adjustments of rivers and develop strategies for mitigating these changes' impacts on ecological systems.

## 6. Conclusions

The findings of this study emphasise the critical role that flood-event variability and sequencing play in shaping the morphological response of fluvial sand and gravel-bed systems in cold regions. The results demonstrated that even in areas where hydroclimatic changes are not yet fully visible, flood-event characteristics are evolving and remain closely linked to specific climatic conditions. Each flood-event type produced distinct morphological responses, such as the formation of riffles and pools during single-peaking floods, and more fragmented and irregular bed forms in double-peaking floods. Additionally, sediment grain size significantly influenced the spatial distribution of erosion and deposition. The increase of double-peaking flood-events, coupled with rising temperatures, could lead to a shift in sediment transport regimes, resulting in heightened geomorphic activity and altered sediment loads. The results underscore the importance of assessing hydroclimatic conditions and flood hydrograph sequences at the catchment scale to accurately predict future morphological adjustment as the impacts of hydroclimatic shift are not uniform across the arctic. Future research should focus on the combined impacts of flood sequences, precipitation patterns, and sediment transport dynamics to develop effective strategies for managing the evolving river systems under climate change. These changes are expected to affect long-term river stability, with significant implications for river ecosystems and flood risk management.

**Data availability**

The climate data is openly available on Finnish Meteorological Institutes (FMI) data service. The Polmak discharge station data is openly available on Norwegian Water Resources and Energy Directorate (NVE) data service. All the other data is available on request.

**Author contribution**

Linnea Blåfield – Writing the manuscript, Field work, Methodology, Formal analysis, Visualisation, Funding.

Carlos Gonzales-Inca – Formal analysis, Editing the manuscript

Petteri Alho – Field work, Data curation, Resources, Reviewing the manuscript, Funding, Supervision

Elina Kasvi – Field work, Reviewing the manuscript, Funding, Supervision

**Declaration of competing interest**

The authors declare that they have no conflict of interest.

**Funding**

This study was funded by the Kone Foundation (202104246), AnthroCliMocs (355018), and by the European Union's Next Generation EU recovery instrument (RRF) through the Research Council of Finland projects: HYDRO-RDI-Network (337279), Green-Digi-Basin (347701), and HYDRO-RI-platform (346161). The study received support also from the Flagship Programme funding granted by the Research Council of Finland for Digital Waters – DIWA Flagship (359247).

**Acknowledgements**

The authors would like to thank research assistant Oona Oksanen from the Fluvial and Coastal Research Group (University of Turku) for helping with the data processing, and other group members who have participated in the field work.

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
