# Peer review of "Morphological response to climate-induced flood-event variability in"

_EGUsphere, 2024_

## Author Response (AR1)

I appreciate the opportunity to review this insightful manuscript, which examines the impact of climate-induced flood variability on the morphological changes of a sub-arctic river. The study addresses a critical issue in river geomorphology, offering valuable insights into how climate change affects sediment transport and river morphology in cold regions. The 32-year dataset and morpho dynamic modeling are significant strengths, providing both observational and computational perspectives on climate-induced changes in river systems.

From my point of view, the manuscript offers valuable and very timely contributions to the field. However, there are areas that could benefit from further refinement. Incorporating recent studies on warming-driven erosion and sediment transport, particularly in permafrost areas, would broaden the context. Additionally, the manuscript would be strengthened by more empirical evidence, such as observable morphological shifts, to support claims regarding sediment transport dynamics during multi-peaking floods. A clearer explanation of the methodology and its limitations would improve the transparency of the analysis. Finally, a deeper discussion on the role of permafrost thaw and riverbank erosion would enhance the manuscript's relevance to current hydrological and geomorphological research.

Overall, I would recommend a moderate revision.

Response to reviewer:

Thanks for the review, we have modified the manuscript based on your comments and suggestions. We have added recent findings on cold climate sediment transport in seasonally frozen ground to the introduction. We don't want to go too deep in permafrost dynamics since this river studied is not a permafrost river. Unfortunately, we don't have long time-series of empirical evidence on migration rates, but we have added stronger justification based on previous studies done in the same region with findings which support our claims. In addition, we have modified the methodology section based on your comments and suggestions to make it more see-through. We have added discussion about seasonally frozen ground and freeze-thaw dynamics on the discussion section. We discuss shortly about permafrost in global scale, however, we do not want to address permafrost thaw too much in this section since it is not relevant for this study site. Hopefully our modifications made based on your comments have improved the manuscript.

**Major Comments:**

Lines 40-47:

The introduction and discussion provide a solid overview of the impact of climate change on river morphology. However, I believe it would enhance the manuscript to compare with recent studies addressing warming-driven erosion and sediment transport in wider cold regions in a more detailed way. This could place the study in a broader context, providing a more comprehensive framework and thus potentially broadening its appeal to a wider audience. Many sub-arctic rivers drain through frozen landscapes. I also wonder whether the catchment is a catchment with permafrost and seasonally frozen ground and this aspect should be better introduced in the introduction. Please check the permafrost map (https://www.sciencedirect.com/science/article/pii/S0012825218305907)

and add such information in the study area Figure 1. Also, some new progress for permafrost river dynamics under climate change are:
https://agupubs.onlinelibrary.wiley.com/doi/10.1029/2024GL112752;
https://agupubs.onlinelibrary.wiley.com/doi/10.1029/2024GL111536;

Response: Thanks for the comment. We have added more details on warming-driven erosion impacts on river morphology, sediment transport and migration rates to the introduction section, rows 50-74. In the northernmost Finland, we have very limited amount of sporadic permafrost. Small patches can be found, mostly in a form of Palsa mires, and from fell summits above the treeline (~400m amsl) where in some cases the bedrock is permanently below 0 degrees. This catchment/river network studied in this paper does not have permafrost to our knowledge based on the research conducted in the area during the past 20-years. However, we added a map of the potential permafrost areas (10-50 % probability) and Palsa mires to figure 1, based on the Nordic permafrost map of Gisnås et al. (2017) (https://doi.org/10.1002/ppp.1922). The lack of permafrost in Finland is due to the warming effect of the Gulf Stream and North Atlantic Drift, limited high elevation areas, thick snow insulating the ground during winter, and abundance of wetlands with warm waterlogged soil and groundwater flow. In Gisnås et al., (2017) the figure 12 shows the modelled distribution of sporadic and discontinuous permafrost between 1980-2010, and indicates that in the region studied in this current study, there is no discontinuous permafrost, and hardly any sporadic permafrost (with very low probability) left this day. There are no permafrost findings from the field either.Therefore, we do not consider this catchment/river network as permafrost river, even though it is located in subarctic region. The ground/soil, however is seasonally frozen during winters which affects the erodibility of river banks during spring flood (https://doi.org/10.1002/esp.4796 and https://doi.org/10.5194/egusphere-egu24-10175 and https://doi.org/10.1002/2013WR014106). We now mention this on introduction and study area description as well (rows 104 and 130).

Lines 440-450:

In the " 5.2. Flood event types and morphological response" section, I believe it could benefit from an explicit reference to permafrost dynamics. The thawing of permafrost significantly impacts riverbank stability, which in turn can alter sediment availability and transport processes. This factor is absent from the manuscript. Additionally, the discussion of future morphological changes mainly emphasizes increased sediment loads due to hydroclimatic shifts, but it would be important to also consider potential changes in riverbank erosion and meander migration rates, which are highly relevant in the context of permafrost thaw and sediment transport dynamics.

Response: Thanks for the comment, we do not consider this river network as permafrost river and therefore we have not addressed permafrost dynamics. However, based on your comment we have added reference to freeze-thaw dynamics of seasonally frozen ground, and how that impacts bank erosion/sediment transport volumes. In addition, we added wider discussion about bank erosion, migration rates and sediment transport dynamics in

context of freeze-thaw dynamics related to seasonally frozen ground to this section. Rows: 475-485.

Lines 452-462:

The study suggests that the increasing frequency of multi-peaking floods could lead to long-term shifts in sediment transport regimes, potentially destabilizing the channel. While this is a valuable observation, the evidence provided seems to be inferred rather than directly demonstrated. It would greatly strengthen the argument to present evidence of observable morphological shifts in the study reach over the 32-year period. For instance, a comparison of historical channel adjustments (e.g., planform changes, bank erosion rates from in-situ or remote sensing observations) would provide empirical support for the claim of long-term changes in river morphology due to the increasing frequency of multi-peaking floods.

Response: Thanks for the comment, this river is relatively narrow and accessing migration time-series from remote-sensing observations (satellite images) is basically impossible, since you can't spot the river from the images. National Land Survey of Finland has aerial images from the area taken in years 1961, 1993, 2004 and 2015 (https://kartta.paikkatietoikkuna.fi/?zoomLevel=9&coord=539507.7631314445_7757882.13 5039186&mapLayers=801+100+default,3400+100+×eries=1961&uuid=90246d84-3958-fd8c-cb2c-2510cccca1d3&noSavedState=true&showIntro=false) but this time-series is too sparse to analyse trends in meander migration or bank erosion rates. From that time-series of historical aerial images, it is however possible to notice that the migration rates of this river channel are very low (~10-15 metres in ~60 years). No notable changes in planform types can be detected from the historical aerial images. We are currently working on studies focusing on laser scanned bank erosion time-series of ~20-year biannual measurements of this river reach as well as time-series of the morphological planform adjustment, but it is too early to say about the results, whether or not it is possible to identify increase/decrease in morphological activity during that relatively short time period. Therefore, we based our claim that increase in multi -peaking floods could lead to increased geomorphic activity over time, to the findings of this study, findings from previous studies, and findings of morphological and hydroclimatic factors at the same river system, and in other rivers around the subarctic/Arctic region. Previous studies show that this river reach experiences mostly vertical erosion and the lateral changes are of low magnitude (Kasvi et al., 2012 & 2017; Lotsari et al. 2014; Salmela et al., 2020). Annual bank erosion is measured to be from no change at all to max of 0.6m at certain locations, mostly between 0-0.2m (Lotsari et al., 2019). In the same study, it is found that most frequent changes in river banks happen during spring flood peak, where as changes with the greatest magnitude happen during falling limb of the spring flood. Rainfall induces frequent small-scale bank erosion in other seasons. The bank material (cohesion) and whether the bank is frozen or not has significant impact on the erodibility of the bank during spring flood. In addition, observations of melting ground in Siberia have indicated increased bank and valley slumping in a large arctic river (Séjourné et al., 2015). Therefore, bank erosion processes are expected to become even more important for sediment supply, leading to higher annual sediment yields in (presently) subarctic areas. Therefore, we based our claims of likely increasing geomorphic activity leading to significant changes in sediment transport rates and morphological adjustment over time on previous climatic, hydrological and morphological research findings from the same region and similar river systems, as well as our own results which are pointing to that direction. We address this issue in row 530-561.

**Minor Comments:**

 Lines 167-171:

In the "3.2. Hydrograph classification" section, the study classifies flood hydrographs into four distinct categories, but I feel that the rationale for selecting the 75th percentile (p75) as the threshold for flood discharge could be further explained. Why was this specific quantile chosen? It would be valuable to explore whether other quantiles (e.g., the median or the 90th percentile) might result in different classifications and what implications such variations could have on the analysis. Providing a clearer justification for the chosen threshold would enhance the transparency of the methodology.

Response: Thanks for the comment, we selected the 75[th] percentile because the use of 90[th] percentile confined the hydrograph data too much. With p90 only the highest peaks of the hydrographs were detected leaving out the important rising and falling phases when evaluating sediment transport dynamics. In addition, with p90, some years no spring flood could be detected at all as moderate or low spring flood peaks did not reach the p90 value. Therefore, we decided to use p75 to include the rising and falling phases of the flood hydrographs, and to detect flood hydrographs also in years with low and moderate spring flood peaks. This issue is now addressed in rows 190-196.

 Lines 178-184:

While the study classifies flood events based on peak sequencing, it does not address whether these sequences are driven by intrinsic hydrological processes (e.g., soil moisture memory, antecedent conditions) or external climatic factors. A more detailed discussion of the underlying drivers of peak sequencing would add depth to the analysis and potentially strengthen the study's conclusions by clarifying the factors that influence flood event sequences.

Response: Thanks for the comment. The classified events are spring flood events. In this region spring floods are driven by external climatic factors, e.g. temperature rise and rainfall, which cause the snow to melt and river ice-cover to break, leading to high discharge peak. This issue and the affect of adjacent conditions are now addressed in rows 211-219.

Lines 327-335:

The analysis suggests that sediment transport rates during the second peak of multi-peaking events are lower than during the first peak, which is consistent with previous findings on sediment depletion. Nevertheless, it would be valuable to consider whether there is any evidence of hysteresis reversal due to finer sediment contributions. If possible, separating the suspended sediment and bedload data in the analysis could provide a more comprehensive understanding of the sediment transport dynamics during multi-peaking events.

Response: Thanks for the comment, unfortunately separating the suspended load from the total transported sediments (TTS) does not provide the information of hysteresis reversal, possibly caused by finer sediment in this case, as the modelled sediment fractions were generally too large to be transported as suspended load. This river system has very low 0-

180mg/l suspended load during flooding, thus we did not value modelling the smallest sediment fractions even though the van Rijn's equations considers both, bedload and suspended load. Thus, the amount of suspended load in the model is low, and we cannot separate the different grain sizes from the bedload. However, previous studies have found that hysteresis reversal can be due to bank erosion, which the model did consider. Previous studies (Lotsari et al., 2014; 2024; Yang et al., 2024) found that bank erosion intensifies during the falling limb of the flood hydrograph, thus this could explain the hysteresis reversal. Reversal could be therefore explained by bank erosion contributing to the sediment flux between the peaks and during the rising limb of second peak in event D. We have now addressed this issue in rows 478-492.

Figures 1-9:

Some of the figures would benefit from clearer labeling, particularly in the distribution of climate data and the identification of flood event types. Additionally, ensuring that the legends and axis labels are consistent across the figures would enhance clarity and facilitate easier comparison of the results.

Response: Thanks for the comment, we have modified legends, labels/axis's from figure 1, 4, 6, and 8 to make them clear and consistent.

This is a nicely written paper merging field data and morphodynamic modeling to understand how (a) how climatic changes modify flood characteristics and [by consequence] (b) how changing flood characteristics impact the morphological response to floods of a sub-artic river. The stated aims of the study are to:

i) Analyse and classify the variation in flood event hydrographs over the past 32 years in a sub-arctic river

ii) Link the flood events to seasonal and annual climate conditions, and

iii) Evaluate the channels morphological response distinctive to each flood event type utilising morphodynamic modelling and sediment hysteresis analysis

These aims are obviously very important given the rapid environmental changes occurring in sub-arctic environments, and I think the authors have nicely achieved their stated aims in the paper. However, there are several aspects of the paper which I think are slightly lacking. For this reason I suggest that the work would be excellent for publication in ESurf after some minor/moderate revisions centered around the following two concerns:

**Main comments:**

First, the presentation of the morphodynamic modelling results is a little bit "black and white", in that the authors do just four simulations with a set of parameters taken from their group's earlier papers, then they discuss the outcomes of these simulations in a definitive <> type of way, when in fact, the input parameters of these simulations are uncertain and the outputs will definitely depend on them. At the same time, the morphodynamic modelling must lack processes contained in the real world (sediment transport fluctuations?, realistic width changes?, vertical sorting? 3D flow effects? frozen vs thawed banks? within-channel ice?), although the potential lack of any such processes is not acknowledged in the methods or discussion. The reader may naturally be curious as to how robust the authors' modeling conclusions are on erosion and deposition patterns (Figure 9), net sediment budget across hydrographs (Paragraph at 359), stream power trends, and hysteresis patterns for the different hydrograph types they define (Figure 8), and other things. Clarifying the sensitivity of the simulation results on the inputs and model assumptions might be possible with an additional paragraph in the results. The limitations of the modeling approaches might be  described in additional text in the methods or discussion.

Thanks for the comment, we have worked on this in the manuscript and now present the uncertainties and limitations related to the modelling conducted in this study. It is true that all the simulation results depend on the parametrisation which of course is simplified compared to the complex dynamics of real world. In Delft3D the sediment input fluctuation at the model input boundary is solved based on the Neumann law. This is of course also an estimation based on the transport dynamics and volume within the model. The sediment transport is updated at each time step based on the flow conditions, and the source and sink terms of the model. This of course do not match the real world spatiotemporal fluctuations in sediment transport but the volume of sediment in transport at each mesh cell (2x2m) is updated at every timestep. Therefore, spatiotemporal fluctuation in sediment transport is considered in each model cell at each times-step, this of course does not match the real world fluctuation. We have now modified the text to less definitive type and clarified the sensitivities in modelling section and discussion. Hopefully these changes made the modelling part more see through like you requested.

The second comment concerns section 4.1 in the results. As the reader arrives at this section, you have previously defined the four different flood-event types: high one-peak floods, low one-peak floods, two separate peaks, and wavy peaked floods. The sorting of your observed hydrographs into these four categories is nicely laid out in Figure 3, where the "typical" hydrographs of each type are shown as red lines. However, the climatic data show only rather weak differences across categories in almost all cases. The box-plots in Figure 6 show that climatic variables are similar (overlapping) across event types. This overlapping makes it tough to believe the claims in 260-272 at face value. With more or less data, we could easily imagine making other conclusions. Can the authors do something more quantitative to strengthen their claims despite the necessarily limited sample size? For example, "High annual and low springtime precipitation were linked with high peak floods". "Wavy flood events (D) experienced the warmest temperatures, high amount of snow, 270 and high levels of both, annual and spring precipitation (Fig. 6)" What do you have for the reader who sees four overlapping box plots which show no significant distinction? In my view this is a pivotal issue since one of your key points is that "each flood event type could be linked to slightly different climate conditions", while it's not immediately obvious that this is the case looking at Figure 6. I would suggest some additional statistical analyses could be added (maybe ANOVA to find significant differences with a given confidence?) to strengthen the claims near lines 260-272.

Thank you for the comment. It is true that with more or less data, different conclusions could be drawn. We also acknowledge that we excluded certain variables—such as solar radiation, snow-water equivalent, ground frost, etc.—which play important roles in snowmelt, runoff generation and volume, and infiltration. Our intention was to keep the analysis focused and to concentrate on the primary climatic drivers. Additionally, many of the excluded parameters lacked sufficiently long or consistent time series, which limited their usability.

In response to your suggestion, we have now conducted ANOVA on the available variables and revised Section 4.1 accordingly. To gain more detailed insights, we also divided the climatic variables into a broader set of sub-groups. Whereas we previously focused only on annual and spring conditions, we now include cold season conditions (October–May) and May-specific weather, which typically coincides with the main melt period.

These additions have improved the resolution of our analysis and provide a more comprehensive understanding of the climatic factors influencing flood-event types. While not all variables showed statistically significant differences between event types, the ANOVA did reveal new findings that help to support our interpretations. We believe the updated results now offer a stronger foundation for our key claims.

**Secondary comments:**

- 47 run-on sentence

Thanks for the comment, this is now fixed. Row 54

- 54 run on sentence

Thanks for the comment, this is now fixed. Row 59

- 60 awkward sentence

Thanks for the comment, this is now fixed. Row 65

- 65 Suggest an edit of this sentence, such as "... through analysis of sediment transport hysteresis patterns, which reflect..." as it's currently a bit awkward seeming

Thanks for the comment, this is now fixed. Row 89

- 71 run on sentence

Thanks for the comment, this is now fixed. Row 94

- 74 You alternate between using oxford commas and not using them. You might aim for consistency in this.

Thanks for the comment, this is now fixed throughout the paper.

- 89 It may be useful to share the bankful width of the river to give more sense of scale.

Thanks for the comment, this is now fixed. Row 125

- 93 run on sentence

Thanks for the comment, this is now fixed. Row 136

- 96 and both the Atlantic ocean and ...

Thanks for the comment, this is now fixed. Row 138

- 100 you never use the abbreviation "a.s.m.l." again, so I guess there is no need to define it.

Thanks for the comment, this is now fixed. Row 124

- Figure 1 - it is not immediately clear to me what "1, 2, 3" marked on the map mean, although it comes together when I stare at the figure a bit. You might add explanation of these numbers in the figure caption

Thanks for the comment, we have modified the figure 1 based on your comments about the labelling, and the other reviewers comments related to permafrost areas.

- 123 I am wondering at this point exactly how often the sediment transport samples were collected

Thanks for the comment, this is now described in detail in the sediment sampling section 3.4

- 128 This should be a colon, not a semicolon: "... based on a combination of field data were generated: ..."

Thanks for the comment, this is now fixed. Row 168

- 133 intervals

Thanks for the comment, this is now fixed. Row 174

- Fig 2A - "WL m" I infer to be "water level" but this is not defined - the figure caption calls it "x". Suggest to say "Regression curve between discharge measurements (Q) and LeveLogger water level (WL) in ...". Also I noticed at 381 you refer to "x" and "y", which I guess is clear enough as "horizontal axis" and "vertical axis", but it may be better to just say the name of the variable you're speaking about.

Thanks for the comment, this is now fixed in the figure caption.

- Fig 2B - the regression made between Pulmanki and Tana looks overfitted, since it decreases at large discharges. Why would the discharge in Pulmanki decrease when Q increases beyond 2200cms in Tana?

Thanks for the comment. The Pulmanki river drains a small catchment of low elevation at the downstream of Tana. The Tana river has a ~20 times larger catchment compared to Pulmanki, and it has high elevation areas reaching over 1km above the sea level. This is the reason that Tana river discharge remains high or still increases when Pulmanki river has already drained its catchment and the Q starts to decrease. Pulmanki region is usually already snow free when there is still snow masses on the upstream and on high elevation areas in Tana catchment. This is the reason why Tana discharge remains high when Pulmanki discharge starts to decrease.

- Table 1 - several quantities here are not defined in the paper. What are MAE or SDE? What is "n" - Manning? The notation of R-squared is typically written `R^2`, with a superscript. Is r a Pearson correlation coefficient or is it \sqrt{R^2}. A definition is needed for these

Thanks for the comment, we have now defined these quantities in the table caption as requested.

- 168 - Figure 6 shows also April floods lumped in with may and june. Should you say April, May, and June here?

Thanks for the comment, we have now fixed this as requested on the figure caption and in the text with more general "Spring floods". Row 207

- 172 - it is natural to wonder how your analyses would vary with changes to your definition of 40cms for a high flow or to your savgol filter parameters. This is a similar-in-spirit comment to the major comment #1 above. In addition, your analyses would not be completely reproducible without details of how you set the peak-finding parameters (prominence, width, etc) in scipy.signal.find_peaks. I think a few sentences about sensitivity and further explaining your peak finding strategy would improve the reader's confidence in your results and enhance reproducibility.

Thanks for the comment. We appreciate this important observation and have corrected it. Now in the methodological part of the manuscript (3.2. Hydrograph classification, line 204) we have added the peak finding parameters as suggested. We also performed sensitivity analyses using different parameter values. The peak finding and classification is mostly influenced by the selection of low threshold values to identify the number of flood events, namely the 50, 60, 70, 80, and 90 percentiles as threshold values. Although variation in the number of identified peaks was observed, particularly

in the 50 and 90 percentiles. However, using threshold values of 70 and 80 percentiles was observed to capture most of the relevant peak flood events. Therefore 75 percentiles seem to be an adequate threshold for this study. Rows 205-218.

- 181 I have seen up to now many examples in the paper where compound adjectives are not hyphenated, e.g. "precipitation-driven discharge peaks", "high-lattitude rivers", "flood-event shapes", "one-peak flood", on and on. I would suggest to search the paper for compound adjectives and add hyphens everywhere applicable, as this makes the reading smoother and reduces any chance of misunderstanding.

Thanks for the comment, these are now fixed throughout the paper.

- 195 were selected

Thanks for the comment, this is now fixed. Row 251

- 198 precipitation event magnitudes? Or occurrence of precipitation events?

Thanks for the comment, we were talking about event magnitudes and this is now made clear on row 255

- 206 to identify

Thanks for the comment, this is now fixed. Row 262

- 207 recurrence or occurrence?

Thanks for the comment, we meant occurrence as in the figure 7. This is now fixed on row 263

- 209 MK is defined here but not M-K strictly speaking

Thanks for the comment, this is now fixed. Row 261

- 209 error in citation formatting

Thanks for the comment, this is now fixed. Row

- 211 removed or compensated for, not neglected I think

Thanks for the comment, this is now changed to "removed". Row 265

- 220 here I am still wondering about the frequency of sediment sampling and how this compares to your detailed dataset on hydrological variables

Thanks for the comment, this is now clearly explained in this section. Row: 276-284 "A total of 70 grab samples (ca. 500 g) and 24 bedload transport samples were collected during various discharges from the area of interest. Grab samples were collected across the entire 6-kilometre reach during a single autumn field campaign under low discharge (4.2 m³/s) conditions. Samples were taken from

the channel bed at left and right bank of each meander inlet, apex and outlet. Bedload transport samples were obtained during both, spring and autumn campaigns, under varying discharge levels (7.5 m³/s, 56 m³/s, and 4.2 m³/s). Twelve bedload transport samples were collected per campaign, each with a sampling duration of six minutes"

- 220 intervals

Thanks for the comment, this is now fixed. Row 284

- 221 the gradistat program, the method of moments, a logarithmic distribution (missing articles "a" and "the", I also have seen other places in the paper with this small error)

Thanks for the comment, this is now fixed on row 286, and we have fixed other similar errors throughout the paper.

- 236 model's geometry (apostrophe in wrong place)

Thanks for the comment, this is now fixed. Row 302

- 237 run-on sentence

Thanks for the comment, this is now fixed. Row 302

- Table 2 - suggest to format m3 as a superscript m^3

Thanks for the comment, this is now fixed. Row 338

- 254 Section, not chapter

Thanks for the comment, this is now fixed. Row 339

- 255 from the field

Thanks for the comment, this is now fixed. Row 340

- 260 it's confusing how you speak of "flood events A-D" in this particular section. You use singular tenses, as in "The wavy event D had an average duration of 9 days" which suggests you're using the typical events, i.e. the Red curves from Figure 3, but rather you are using all events of a particular type A-D in this particular subsection. I suggest to speak of "flood events of type A" and so on, in a plural tense. It should be clear you're discussing the statistical outcomes of many floods. Another example - you say "it had more variability than type B", but you really mean "the flood events of type A had more variability than those of type B" and so on.

Thanks for the comment, this is now fixed throughout the paper.

- 267 what type of preciptation amounts?

Thanks for the comment, it was supposed to say moderate. However, this section is now re-written after the ANOVA, and now we have specified it as "rainfall in May". Row 357

- 268 it -> flood events with two separate peaks

Thanks for the comment, this section was re-written roe 359 onwards.

- 277 "In general, there were more variation in spring variables than annual variables, which implicates that the hydroclimatic conditions preceding the spring flood impact the flood event type more than the prevailing spring conditions." - Are you sure? Could the larger variation in spring variables not be simply that the spring sample size is smaller? (implicates -> implies)

Thanks for the comment, we have run more statistical analysis using ANOVA like you suggested in the main comments and this section is now refined based on the new results. We do acknowledge that the sample size has an impact on the results and we now discuss this in the discussion section. However, we feel that we can now more confidently say that the proceeding annual condition have an impact after the ANOVA results confirmed that. Implicates is now corrected as suggested.

- 334 and in Figure 8. The linkage between hysteresis type and hydrograph shape seems to me to be a very nice result that is worth emphasising. Currently this is not mentioned in the abstract or key points, although "hysteresis" is in the keywords.

Thanks for the comment, we have now emphasised this in the abstract and key points, results and discussion section throughout the manuscript as it was indeed left with a less emphasis in the previous version.

- 426 you refer in the paper to "sediment hysteresis" but it's not hysteresis of the sediment, it's hysteresis of the sediment transport. I suggest to modify everywhere to "sediment transport hysteresis"

Thanks for the comment, this is now fixed throughout the paper.

- Figure 9 shows beautiful patterns of erosion and deposition in your morphodynamics simulations of the study reach. I guess it would be possible to make videos of the channel evolution across the four hydrograph types rather easily. This would make a nice addition to the paper as supplementary info which would increase its visual appeal and show how the erosion/deposition and sediment export differences between the four event types you've defined actually arise. I would suggest if it's easy enough, you might make these videos and integrate them into the text with a few sentences of discussion. This would strengthen the paper

Thanks for the comment. We agree that adding videos would be a great addition to the manuscript. However, producing them to match the style of Figure 9, including consistent coloring and channel confinement, would require considerable effort. We would need to extract the geometry for each timestep from each model run and process them individually in a GIS environment. While the model software does include a built-in animation tool, it's quite limited. The mesh is always visible, and there's no way to isolate the channel from the surrounding mesh. As a result, the finer morphological details get lost, especially since the channel appears very narrow within the much wider mesh area. The proportions of the modelled area are so unbalanced (long and narrow) that it's difficult to interpret the morphological details from aanimation without additional GIS-based processing.Therefore, we decided to stick with the current presentation of the modelled final geometry.

---

## Author Response (AR2)

Response to the Editor

The changes and additions requested are marked to the manuscript with red font.

28: You could provide more details of how hydrograph shape affects the geomorphic response.

Response: Thanks for the comment, we added more details "Double-peaking floods resulted in relatively more heterogeneous and complex morphological outcome compared to single-peaking floods"

63: You could explain why the order and duration of flood peaks is important.

Response: Added a sentence " In multi-peaking floods, the order and duration of different peaks significantly affects the sediment transport volume and the pattern of sediment transport hysteresis because the flow conditions are controlling when, how much, and what type of sediment is mobilised, reworked, or deposited within the river system (Mao, 2018)."

164 and 276: In both places, say what years the sediment samples were collected from.

 Response: Added the year to both lines

187: It would be helpful to include the comment in your response document about why the Pulmanki discharge might go down as the Tana discharge goes up, as readers might have the same question.

Response: Added explanation to the text: "The Tana River discharge continues rising even though the Pulmanki River discharge decreases due to the fact that the Tana River drains 20 times larger catchment compared to the Pulmanki River. The Pulmanki River catchment (sub-catchment of Tana) is located on the downstream of the Tana River and has much lower topography than the Tana catchment, which drains areas up to 1,1 km from the sea level. This causes delay in the snowmelt and the runoff peaks occur later compared to the Pulmanki catchment."

293: Is it surprising that bedload looks coarser than grab samples - is this because samples come from different years maybe?

 Response: The samples are collected during spring and autumn flow conditions in 2019. The bed load samples are finer than the grab samples (100 microns = 0.1 mm, 1000 microns = 1mm). This can be Helley-Smith error as it has tendency to "vacuum" in the smaller grain sizes.

393: It's not clear what you mean by occurrence - date of the year when that event occurred, or just whether it occurred in that year? If you were just looking at whether it occurred in that year, then what statistics did you use – can you apply a M-K test to a dataset of ones and zeros?

Response: M-K test can be problematic when applied to binary data since this kind of data does not have a gradual trend. That's why we first calculated the intervals between the annual occurrence (number of timesteps between the events). This way we could apply M-K test to see if the spacing interval between the events is increasing or decreasing over time.

419: Specify somewhere that TTS is calculated across the entire model, rather than being at a single x-section.

Response: We now mention that the TTS was calculated across the entire model area.

488: Define geomorphic activity. Why is it plotted as a pie chart?

Response: Geomorphic activity was calculated based on the amount of total mobilized sediment within the active area (TTS divided by the active channel area in the model). Pie chart is probably not the best choice to plot it since the values are individual instead of percentages of whole. However, we felt that it is easier to separate the charts from each other in the figure if the charts describing different factors are different types.

513: Can you add reference to a figure that shows that type B floods increased in volume?

Response: Added reference to Figure 7.

584: In this paragraph there are lots of sentences starting with just 'this' - try and avoid starting sentences with 'this' as there is a risk that the reader does not know what 'this' refers to.

Response: Thanks for the comment, we have now modified the chapter to avoid using "this".

590: Not clear to me if the higher fine sediment from bank erosion is a possible explanation for the model results, or what you would expect to see in the field. Clarify whether the model included bank erosion.

Response: In the section " 3.4. Morphodynamic modelling " we tell that "The default scheme for dry-cell erosion of banks was applied without further adjustment, as the focus of the study was on longitudinal sediment transport and vertical changes to the channel bed". So yes, bank erosion was included with the default values. So that's the reason we consider that bank erosion might be possible contributor to the fast response of TTS and the hysteresis pattern.

556: I can see why this has been added in response to the reviewers. However, it would be good to also consider uncertainty as specific to this study. For example, if you have selected a different flood to be characteristic of each of the four flood types, do you think that you still would have seen the same patterns between the flood types? i.e. how much variability would you expect to see between floods of the same type as compared to between the different groups?

Response: Thanks for the note, we have added discussion about the issue you have raised as follows "In addition, only one hydrograph was modelled for each flood-event type, meaning that choosing a different hydrograph of the same type could have resulted in different volumetric changes and amount of total transported sediment, as these depend on the flood's volume and transport capacity. However, the findings of this study, along with previous laboratory (Mao, 2012; 2018) and modelling studies (Kasvi et al., 2015), support the view that the shape and sequence of the hydrograph play a crucial role in determining morphological outcomes. Thus, we can expect that by selecting different hydrograph of each flood-event type, we would have resulted to similar type of morphological patterns relative to the magnitude and transport capacity of the flood even though quantitative outcomes might have differed."

Additional private note (visible to authors and reviewers only): Apologies for the time it's taken me to turn this around!